# The transcriptional and phenotypic characteristics that define alveolar macrophage subsets in acute hypoxemic respiratory failure

Eric D. Morrell [1,6] ✉, Sarah E. Holton [1,6], Matthew Lawrance[2], Marika Orlov [3], Zoie Franklin[2], Mallorie A. Mitchem[2], Hannah DeBerg[2], Vivian H. Gersuk [2], Ashley Garay[1], Elizabeth Barnes[1], Ted Liu[1], Ithan D. Peltan[4], Angela Rogers [5], Steven Ziegler [2], Mark M. Wurfel[1] & Carmen Mikacenic [2] ✉

The transcriptional and phenotypic characteristics that define alveolar monocyte and macrophage subsets in acute hypoxemic respiratory failure (AHRF) are poorly understood. Here, we apply CITE-seq (single-cell RNA-sequencing and cell-surface protein quantification) to bronchoalveolar lavage and blood specimens longitudinally collected from participants with AHRF to identify alveolar myeloid subsets, and then validate their identity in an external cohort using flow cytometry. We identify alveolar myeloid subsets with transcriptional profiles that differ from other lung diseases as well as several subsets with similar transcriptional profiles as reported in healthy participants (Metallothionein) or patients with COVID-19 (CD163/LGMN). We use information from CITE-seq to determine cell-surface proteins that distinguish transcriptional subsets (CD14, CD163, CD123, CD71, CD48, CD86 and CD44). In the external cohort, we find a higher proportion of CD163/LGMN alveolar macrophages are associated with mortality in AHRF. We report a parsimonious set of cell-surface proteins that distinguish alveolar myeloid subsets using scalable approaches that can be applied to clinical cohorts.

Alveolar monocytes and macrophages play an essential role in almost all aspects of lung health and disease[1,2]. The majority of macrophages in the healthy human lung are classified as mature CD206+ alveolar macrophages[3–5]. Mature alveolar macrophages are primarily responsible for processing surfactant, patrolling the airspaces for pathogens, and removing dead cells and debris. However, there is strong evidence that significant heterogeneity exists within the CD206+ alveolar macrophage population. For example, mature CD206+ alveolar macrophages collected from participants with idiopathic pulmonary fibrosis have much higher expression of profibrotic genes such as *APOE* and *MMP12* compared with CD206+ alveolar macrophages from healthy donors[6]. In highly inflammatory disease states such as bacterial infection or acute respiratory distress syndrome (ARDS), CD14+ peripheral blood monocytes are recruited to the lung and become alveolar monocytes[7]. Newly recruited alveolar monocytes secrete inflammatory cytokines and chemokines as well as activate T effector cells. Over the past decade, studies using single-cell RNA-sequencing (scRNA-seq) have identified an even wider spectrum of subsets in both

[1]Division of Pulmonary, Critical Care, and Sleep Medicine, University of Washington, Seattle, WA, USA. [2]Translational Immunology, Benaroya Research Institute, Seattle, WA, USA. [3]Division of Pulmonary Sciences and Critical Care Medicine, University of Colorado, Aurora, CO, USA. [4]Division of Pulmonary and Critical Care Medicine, Intermountain Health, Murray, UT, USA. [5]Division of Pulmonary and Critical Care, Stanford University, Stanford, CA, USA. [6]These authors contributed equally: Eric D. Morrell, Sarah E. Holton. ✉e-mail: edmorrel@uw.edu; cmikacenic@benaroyaresearch.org

health[8] and disease[9–20]. The diverse activation states of alveolar monocytes and macrophages in these different diseases are thought to contribute to multiple processes in the lung including inciting injury or promoting repair[1,2].

There are several important knowledge gaps in our understanding of the role that alveolar monocytes and macrophages play in acute lung injury and repair. First, the diversity of alveolar monocytes/macrophages in patients with non-COVID-19 associated acute hypoxemic respiratory failure (AHRF) has not been well-described. It is not known whether alveolar monocyte/macrophage subsets that have been identified by scRNA-seq in participants with COVID-19[9–13] are conserved in critically-ill patients without viral infection. Second, the cell-surface protein markers that best distinguish specific alveolar monocyte/macrophage transcriptional subsets have not been reported. The identification of a parsimonious panel of cell-surface protein markers that accurately identify different alveolar monocyte/macrophage transcriptional subsets would facilitate larger-scale clinical studies that establish robust links between each subset and clinical outcomes. Third, the evolution of alveolar monocyte and macrophage subsets over time is not known. How alveolar monocyte and macrophage subsets transition over time, and the role that peripheral blood monocyte recruitment plays in this transition, is critically important to understanding disease pathogenesis.

In this work, we aim to address these knowledge gaps by applying cellular indexing of transcriptomes and epitopes (CITE-seq) to alveolar cells serially-collected from bronchoalveolar lavage (BAL) and paired peripheral blood mononuclear cells (PBMCs) sampled from a prospective cohort of mechanically ventilated patients with AHRF. CITE-seq is a single-cell approach that combines scRNA-seq with cell-surface protein quantification[21,22]. We determine that the alveolar microenvironment in AHRF is populated by a heterogenous collection of alveolar monocyte/macrophage subsets with specific homeostatic, inflammatory, and reparative transcriptional signatures that are correlated with soluble alveolar mediator levels. Importantly, we leverage CITE-seq technology to identify the cell-surface proteins that distinguish alveolar monocyte/macrophage transcriptional subsets. Finally, we show that the proportion of these subsets is highly dynamic, with the percentages of distinct alveolar monocyte/macrophage subsets either increasing, decreasing, or staying the same over the course of days. Our findings define the transcriptional and phenotypic characteristics of alveolar monocyte/macrophage subsets in AHRF and provide a framework for future studies that seek to isolate or identify specific subsets for functional evaluation and clinical risk stratification.

## Results
### Study population
We recruited critically ill participants from multiple intensive care units at Harborview Medical Center (HMC) in Seattle, WA. Participants were included if they had a risk factor for ARDS (pneumonia, sepsis, trauma, aspiration, massive transfusion, or lung contusion), were supported on invasive mechanical ventilation for < 7 days, had an infiltrate on radiographic imaging, and required supplemental oxygen. Patients with COVID-19 or an existing chronic pulmonary disease were excluded. Full inclusion and exclusion criteria are shown in Table S1. We performed research bronchoscopies and collected paired samples of BALF and PBMCs at the time of enrollment (Bronchoscopy = B1) and again 4 days later (Bronchoscopy 2 = B2) if a participant was still alive and on invasive mechanical ventilation.

Table 1 displays the clinical characteristics of enrolled participants (CITE-seq Cohort). Patients most commonly suffered from severe trauma, were sampled a median of 4 days after initiation of mechanical ventilation, had a median $P_aO_2/F_iO_2$ (P/F) ratio of 198, and 50% met Berlin Criteria for ARDS[23]. We did not perform a B2 on participants who were extubated prior to the sampling window, and no participants died within 2 weeks of B1. The trajectory of hypoxemia, illness severity

(SOFA score)[24], and BAL fluid biomarker levels are shown in Fig. S1. Table S2 displays the amount of blood products transfused into participants prior to B1. All blood transfusion products at HMC undergo a > 4-log reduction of leukocytes according to United States Food and Drug Administration Standards[25], resulting in a negligible amount of blood donor leukocytes transfused into enrolled participants. Cells were cryopreserved at the time of bronchoscopy and thawed in batches on the day of analysis. Prior to CITE-seq analysis, BAL neutrophils were depleted from thawed cells using negative selection with CD66b magnetic beads and cells were sorted for live CD45$^+$CD15$^-$ leukocytes. We generated libraries with an average sequencing depth of 20,000 reads/cell for gene expression and 5000 reads/cell for feature barcodes (cell-surface proteins) (Table S3). We analyzed 64,317 alveolar leukocytes (median, IQR for each participant: 5141, 4800–6171) and 101,866 PBMCs (median, IQR for each participant: 8112, 6762–10,617) from twenty-four total samples (12 BAL and 12 PBMC) after excluding approximately 10% of cells that did not meet quality control thresholds (Table S4, Figs. S2, S3).

### Alveolar myeloid subsets are highly diverse in AHRF
We clustered alveolar leukocytes using CITE-seq data in order to identify subsets present within our cohort of participants with AHRF. This unsupervised analysis identified cell populations that clearly mapped to lineage markers of established cell types such as monocytes, macrophages, dendritic cells (DCs), CD4$^+$ T cells, and CD8$^+$ T cells (Fig. 1A). As anticipated, the most abundant cell types were non-granulocytic alveolar myeloid cells (encompassing alveolar monocytes, alveolar macrophages, and cDCs).

We then did a second round of clustering limited to the alveolar myeloid population. We identified 9 subsets (BAL Clusters 0 – 8) characterized by highly distinct gene expression patterns (Fig. 1B, C, Table S5). Some of the subsets we identified had very similar gene expression patterns to previously reported alveolar macrophage subsets in healthy participants ("Metallothionein Macrophages" – BAL Cluster 8)[8,17], patients with severe COVID-19 ("IFN-Related Macrophages" – BAL Cluster 5)[9–11], or patients with idiopathic pulmonary fibrosis ("Matricellular Macrophages" – BAL Cluster 6)[14,15] (Table S6 summarizes published datasets compared with our dataset). We also identified alveolar macrophage subsets in our participants with non-COVID-19 associated AHRF that are distinct from other diseases such as BAL Cluster 2, which we refer to as "Intermediate Monocyte-Macrophages." Intermediate Monocyte-Macrophages were characterized by high expression of genes associated with mature macrophages such as FBP1 and APOE, but also low expression of other genes that are highly expressed in mature macrophages such as FABP4 and IFI27[8,17]. Fig. 1D displays the percentage of each subset as a proportion of all alveolar myeloid cells at B1 and B2. Our findings demonstrate the alveolar space of patients with varying severity of AHRF is composed of a mixture of alveolar monocyte/macrophage subsets unique to AHRF as well as subsets that are seen in other disease states such as idiopathic pulmonary fibrosis.

We correlated our CITE-seq-derived alveolar myeloid clusters with BAL fluid biomarker levels to explore the relationship between cell subsets and alveolar mediators. The proportions of Mature and Intermediate Monocyte-Macrophages were inversely correlated with the levels of a broad range of proinflammatory and $T_H1$ mediators (Fig. 2A). For example, a higher proportion of Intermediate Monocyte-Macrophages were associated with lower BAL fluid levels of sRAGE (marker of alveolar epithelial injury), CXCL10 ($T_H1$ chemokine), and IL-6 (inflammatory cytokine) (Fig. S4). In contrast, LGMN/CD163, IFN-Related, and DCs clustered together and were positively correlated with proinflammatory chemokines and cytokines such as IL-12b. A higher proportion of Inflammatory Monocytes was strongly associated with higher BAL IL-6 levels. As expected, the proportion of IFN-Related Macrophages was strongly associated with BAL CXCL10 levels.

## Table 1 | Participant Characteristics of the CITE-seq Cohort

| ID | Age* | Sex† | Gender† | Primary risk factor | Secondary risk factors | Sample day | ARDS | SOFA | P/F Ratio | OI |
|---|---|---|---|---|---|---|---|---|---|---|
| 1 | 20–30 | M | Man | Trauma | Contusions/Massive Transfusion | 3 | No | 10 | 357 | 2.5 |
| 2 | 20–30 | F | Woman | Trauma | Contusions/Massive Transfusion | 5 | No | 3 | 468 | 3.2 |
| 3 | 40–50 | M | Man | Trauma | Contusions | 2 | No | 7 | 310 | 3.2 |
| 4 | 40–50 | M | Man | Trauma | Pneumonia/Contusions | 4 | Yes | 9 | 180 | 5.6 |
| 5 | 50–65 | M | Man | Aspiration Pneumonitis | VT/VF Arrest | 3 | Yes | 13 | 215 | 6.0 |
| 6 | 40–50 | M | Man | Trauma | Pneumonia/Massive Transfusion | 5 | Yes | 7 | 180 | 7.2 |
| 7 | 50–65 | M | Man | NSTI/Sepsis | Volume Overload | 4 | No | 9 | 175 | 8.6 |
| 8 | 50–65 | M | Man | Trauma | Pneumonia | 4 | Yes | 8 | 84 | 13.1 |
|  | 47 (32–54) | 87% Male | 87% Man | 75% Trauma | NA | 4 (3–5) | 50% ARDS | 9 (7–10) | 198 (176–345) | 5.8 (3.2–8.3) |

*ARDS* Acute respiratory distress syndrome, adjudicated at time of bronchoscopy, *NSTI* Necrotizing soft tissue infection, *P/F ratio* $P_aO_2/F_iO_2$ ratio; calculated at the time of bronchoscopy (if missing arterial blood gas, SaO2/FiO2 ratio is reported), *OI* Oxygenation index, Sample Day = interval of days between initiation of invasive mechanical ventilation and bronchoscopy; *SOFA* Sequential organ failure assessment (calculated at time of bronchoscopy), *VT/VF* Ventricular tachycardia/ventricular fibrillation

Race and ethnicity were ascertained by participant self-reporting; we do not report individual race/ethnicity to protect participant privacy; 75% of the cohort was white and 25% of the cohort was black

*Age provided as a range to protect participant privacy

†Sex and gender are described based on participant self-reporting

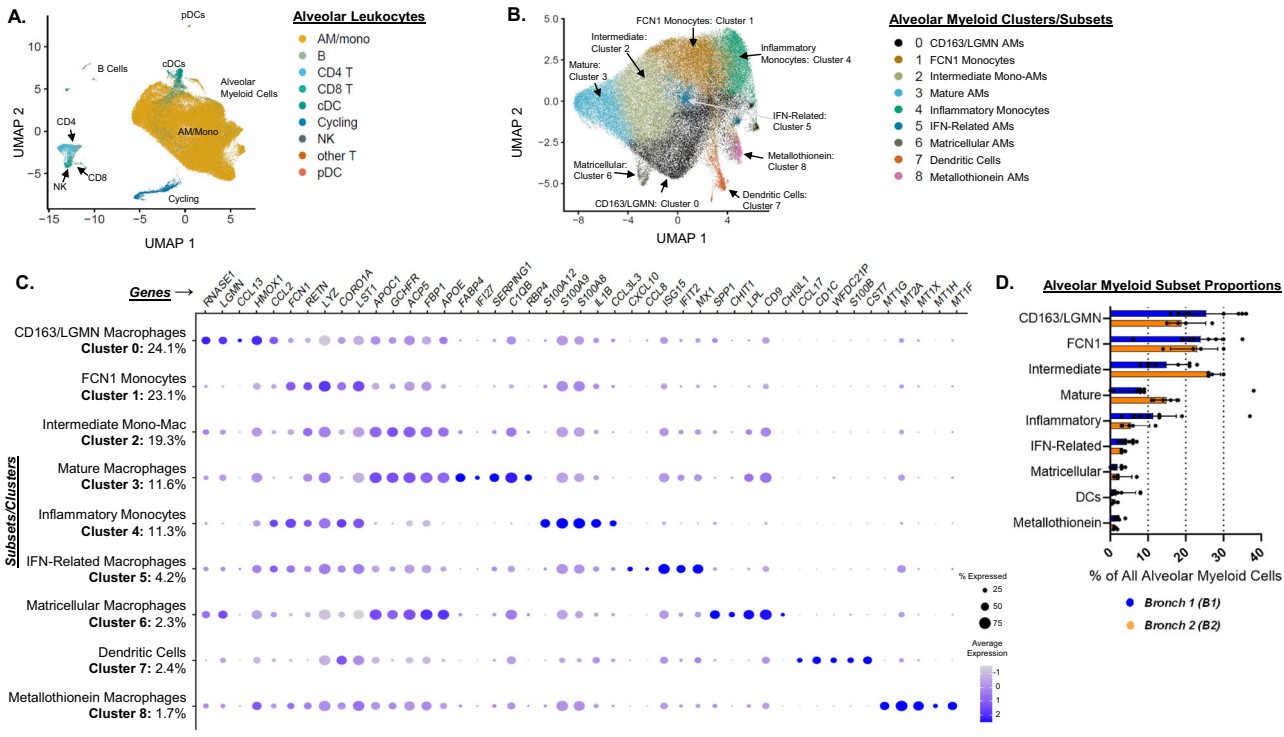

**Fig. 1 | Alveolar monocyte and macrophage subsets are highly diverse in acute hypoxemic respiratory failure.** We performed bronchoscopy with bronchoalveolar lavage (BAL) on 8 individuals with acute hypoxemic respiratory failure supported with invasive mechanical ventilation (Bronchoscopy 1 = B1). Four of the participants were sampled again 4 days later (Bronchoscopy 2 = B2). We isolated single cells and assessed them with CITE-seq. **A** Uniform manifold approximation and projection (UMAP) plot displaying clustering of 64,317 cells based on gene expression. We annotated the clusters by mapping them to published datasets to identify B cell, T cell, myeloid (alveolar macrophages = AM, alveolar monocytes = mono, classical dendritic cells = cDCs), and other cell-types (designated by color).

**B** Cells identified as myeloid (including macrophages, monocytes, and cDCs) in panel A were re-clustered. Color designates assignment of cells to one of the 9 clusters identified by Seurat. **C** Dot plot comparing the expression of marker genes (x-axis) across nine alveolar myeloid cell clusters (y-axis). Each cluster is annotated based on the marker genes (Table S5). The proportion of each cluster as a percentage of all alveolar myeloid cells is displayed. The dot size is proportional the percentage of cells expressing the gene in each color. The color intensity is proportional to the average scaled log-normalized expression within a cluster. **D** Bar plot displaying the individual percentages, median, and interquartile range of each subset as a proportion of all alveolar myeloid cells at B1 and B2.

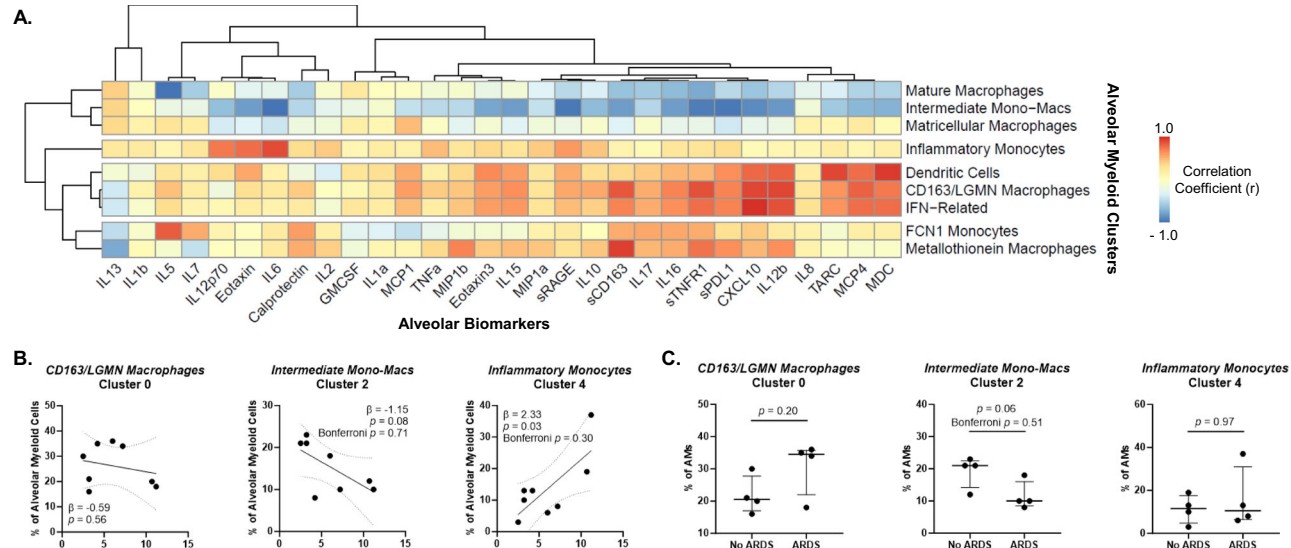

**Fig. 2 | Correlations between alveolar myeloid subsets, biomarker profiles, and clinical severity. A** Heatmap of the correlation coefficients between alveolar myeloid subset proportions (y-axis) and $\log_2$ alveolar biomarker levels (x-axis). Colors represent the correlation with scale indicating value of Pearson's *r* correlation. Axes are ordered by clustering based on Pearson correlation-distances using pheatmap. **B** Associations between the proportion of alveolar myeloid subsets as a percentage of all alveolar myeloid cells (y-axis) and oxygenation index (OI) (x-axis). OI is a measure of respiratory failure severity that accounts for both oxygenation and mean airway pressure being delivered by mechanical ventilation. Higher values indicate more severe respiratory failure. Depicted are the individual values, linear regression best-fit line, and 95% confidence intervals ($n = 8$ unique participants). *P*-values test whether the slope (β-coefficient) is significantly non-zero and are nominal. Bonferroni *p*-values are adjusted for 9 statistical tests (multiple hypothesis testing for an association between each of the nine subsets and the clinical outcome). **C** The percentage of each alveolar myeloid subset as a proportion of all alveolar myeloid cells in participants with or without ARDS. Depicted are the individual values, median, and interquartile range of each subset as a proportion of all alveolar myeloid cells ($n = 8$ unique participants). *P*-values were generated with two-sided Mann-Whitney tests and are nominal. Bonferroni *p*-values are adjusted for 9 statistical tests (multiple hypothesis testing for an association between each of the nine subsets and the clinical outcome).

Collectively, this integrated analysis highlights distinct cell-cytokine profiles reflective of different immune responses in non-COVID-associated AHRF.

We next explored whether the subsets were associated with patient-level factors such as age and severity of respiratory failure to better understand the clinical relevance of the subsets. We detected a nominal association between a higher proportion of Inflammatory Monocytes and worse oxygenation index (Fig. 2B, Fig. S5), however none of the 9 subsets were significantly associated with ARDS, the most severe manifestation of AHRF (Table S7, Fig. 2C). We did not observe significant associations between the proportion of alveolar myeloid subsets and age (Fig. S6), which is consistent with an analysis of six human datasets that did not identify a transcriptionally distinct population of alveolar macrophages that is unique to older individuals[26]. Though preliminary, these results suggest the proportion of alveolar monocyte/macrophage subsets is more closely related to the alveolar microenvironment (e.g. soluble mediators) and clinical severity than intrinsic factors such as age.

**Intermediate monocyte-macrophage subsets are present in the lung**

The contribution of recruited blood monocytes and tissue-resident precursor alveolar macrophages to alveolar myeloid subsets has been evaluated through lineage-tracing experiments in mice[1]. At steady-state, mature macrophages are continually replenished by tissue-resident precursor alveolar macrophages. During injury, blood monocytes are recruited to the alveolar space and mature into alveolar macrophages[27,28]. Despite considerable advances in our understanding of macrophage ontogeny in mice models, the ontogeny of alveolar myeloid subsets in humans is not well-characterized.

We analyzed our data using RNA velocity to better understand the developmental trajectory and fate of human alveolar myeloid subsets.

RNA velocity is derived from the ratio of unspliced (nascent) to fully-spliced (mature) mRNAs in an individual cell[29]. Cells in the steady-state have a low ratio of unspliced-to-spliced mRNA, whereas cells that are transitioning to different developmental states have a high ratio of unspliced-to-spliced mRNA. The developmental trajectory of a collection of individual cells can be inferred by the rate and direction of RNA velocity.

Metallothionein Macrophages and DCs had the shortest median RNA velocities amongst the subsets (Fig. 3A), suggesting these subsets were composed of steady-state or non-transitioning cells. In contrast, Inflammatory Monocytes had the highest median RNA velocity compared with the other subsets. The direction of RNA velocity (projected through partition-based graph abstraction (PAGA))[30] showed that alveolar myeloid subsets originating from either FCN1 Monocytes, CD163/LGMN Macrophages, or Matricellular Macrophages all converge into Intermediate Monocyte-Macrophages (Fig. 3B, Fig. S7). The Intermediate Monocyte-Macrophages were also connected to the Mature Macrophage subset. These findings suggest that heterogenous alveolar monocyte/macrophage subsets converge into a transcriptionally distinct subset such as Intermediate Monocyte-Macrophages over the early course of AHRF. It is possible this subset may repopulate the Mature Macrophage subset or represent a persistently distinct subset. Future studies with longer interval follow-up BAL fluid sampling are required to determine whether cells in this Intermediate Monocyte-Macrophage subset retain their transcriptional programs over time, transition to more mature macrophages, or undergo programmed cell death.

We analyzed paired blood monocytes that were collected at the same time as the alveolar samples to gain insight into blood-lung monocyte chemotaxis. We projected the blood myeloid transcriptional signatures onto the UMAP generated from only alveolar samples (Fig. 3C). Almost all the blood monocytes projected in the same UMAP

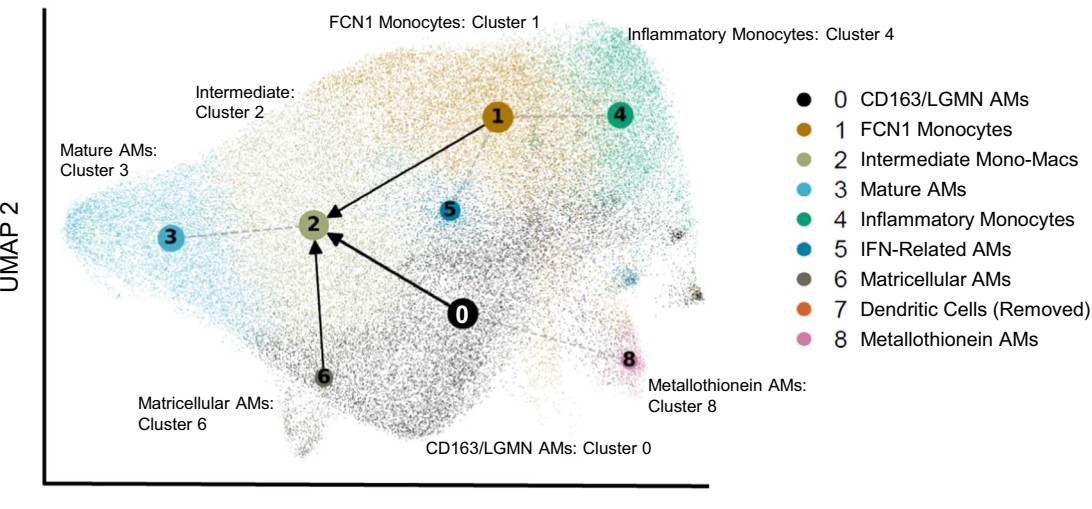

**Fig. 3 | Intermediate monocyte-macrophage subsets are present in the lung.**
**A** Box-plots of median (center line), interquartile range (edge of box), 1.5x inter-
quartile range (whiskers), and individual outliers (dots) of RNA velocity for each
alveolar myeloid subset. **B** Partition-based graph abstraction (PAGA) of RNA
velocity field projected on the alveolar myeloid UMAP (Fig. 1B). Gray dotted lines
represent topologic connectivity of subsets. Arrows represent RNA velocity
trajectory-inference (alveolar macrophage = AM). Dendric cells were excluded
from RNA velocity analysis. **C** We collected paired peripheral blood mononuclear
cells (PBMC) from participants who underwent research bronchoalveolar lavage
(BAL). We isolated single cells and assessed them with CITE-Seq. We selected cells
that mapped to blood myeloid lineage markers (monocytes, macrophages, and
DCs) and then projected them into the BAL UMAP space. Blood monocytes clus-
tered in the upper right of the BAL UMAP (occupying the same BAL UMAP space as
FCN1 Alveolar Monocytes and Inflammatory Alveolar Monocytes). Blood DCs
occupied the same BAL UMAP space as alveolar DCs.

space as the alveolar subsets FCN1 Monocytes, Inflammatory Mono-
cytes, and IFN-Related Macrophages, suggesting these alveolar subsets
may originate from recently recruited blood monocytes. We then
created an "integrated" blood-lung UMAP by combining transcrip-
tional data from blood monocytes and paired BAL samples (Fig. S8,
Table S8). Integrated Clusters 0, 3, and 4 were characterized by
monocyte gene expression whereas Integrated Clusters 1, 2, and 5
resembled macrophages. The proportion of each cluster originating
from a BAL vs. blood sample is shown in Fig. S8C. Notably, non-
classical monocytes (Integrated Cluster 4) were almost exclusively
from blood samples. In contrast, the classical monocyte clusters

(Integrated Clusters 0 and 3) were each derived from ~25% BAL and
~75% blood samples. Our finding that classical monocytes
(CD14$^+$CD16$^-$), but not non-classical monocytes (CD14$^-$CD16$^{++}$), are
present in the alveolar space suggests either selective trafficking of
blood monocyte populations to the alveolar space or that the blood
monocytes are immediately polarized to a classical transcriptional
state as soon as they enter the alveolar environment in early AHRF.

**Cell-surface proteins distinguish transcriptional subsets**
We used information from CITE-seq to identify the cell-surface pro-
teins that best distinguish the alveolar myeloid transcriptional clusters.

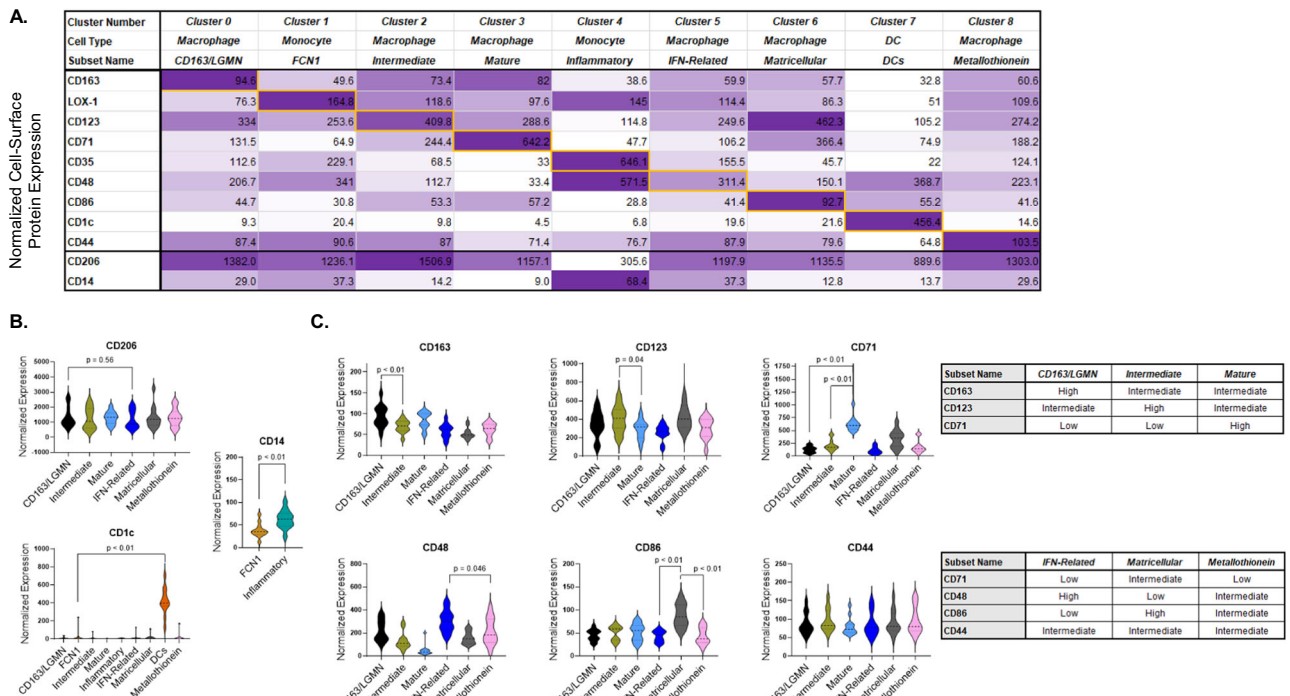

**Fig. 4 | Cell-surface protein markers distinguish alveolar monocyte and macrophage subsets.** We used feature barcodes to identify the cell-surface proteins that best discriminated each alveolar myeloid transcriptional subset. **A** Table displaying the 9 most differentially expressed cell-surface proteins (y-axis) across nine alveolar myeloid cell transcriptional subsets (x-axis). Data on CD206 and CD14 are included at the bottom as a reference. The color intensity is proportional to the average scaled log-normalized expression for each cell-surface protein. Supplementary Data 1 shows the cell-surface protein intensities for each subset. **B** The normalized expression for each cell-surface protein (y-axis) for each transcriptional subset (x-axis). Depicted are violin plots (including median, interquartile range, and 1.5x interquartile range). The p-value was generated with a two-sided T-test of the pair-wise comparison between the two subsets with the largest difference in CD206 expression. **C** The normalized expression for each cell-surface protein (y-axis) for each transcriptional subset (x-axis). Depicted are violin plots (including median, interquartile range, and 1.5x interquartile range). P-values were generated with two-sided T-tests. The tables on the right summarize the relative cell-surface protein expression levels for each alveolar macrophage transcriptional subset.

The most differentially expressed cell-surface proteins amongst the 9 alveolar myeloid subsets are shown in Fig. 4A, Table S9, and Supplementary Data 1. Fig. S9 displays the antigen-specificity scores projected onto the alveolar myeloid subset UMAP. The antigen-specificity score represents the likelihood of a cell-surface protein binding to its receptor compare with the negative isotype controls (and thus accounts for unbound ambient antibodies captured in droplets). CD206 (canonical alveolar macrophage marker) cell-surface protein expression was not significantly different between any of the alveolar macrophage subsets (Fig. 4B). Cell-surface protein expression levels of CD14 (canonical monocyte marker) were significantly different between alveolar monocyte Clusters 1 (FCN Monocytes) and 4 (Inflammatory Monocytes), and CD1c clearly distinguished alveolar DCs from the other alveolar myeloid subsets (Fig. 4B).

Although CD206 did not discriminate between alveolar macrophage subsets, a combination of cell-surface protein levels of CD163 (scavenger receptor), CD123 (IL-3 receptor), and CD71 (transferrin receptor) were able to discern CD163/LGMN, Intermediate, and Mature transcriptional subsets, respectively (Fig. 4C). For example, CD163/LGMN Macrophages were characterized by high CD163 and low CD71 cell-surface protein expression. On the other hand, Mature Macrophages were characterized by intermediate CD163 and very high CD71 cell-surface protein expression. A combination of CD48 (adhesion and co-stimulation protein), CD86 (co-stimulation), and CD44 (cell adhesion and migration protein) were likewise able to discriminate IFN-Related, Matricellular, and Metallothionein Macrophages, respectively. We have identified a parsimonious set of cell-surface protein markers (CD14, CD163, CD123, CD71, CD48, CD86, and CD44) that provides a roadmap for future studies that seek to identify and purify

alveolar monocyte/macrophage subsets from participants enrolled in large clinical cohorts.

### CD163/LGMN macrophages are associated with mortality

We sought to validate if the alveolar macrophage subsets we classified with our CITE-seq data could be identified in an external cohort of critically ill patients, and then determine whether these subsets were associated with clinical outcomes. We analyzed flow cytometry data we previously generated from a cell-surface marker panel (Table S10) applied to BAL fluid samples collected from the HMC Clinical Cohort[5,31]. The HMC Clinical Cohort enrolled critically ill participants undergoing bronchoscopy for suspicion of ventilator-associated pneumonia. Table 2 displays the clinical characteristics of the HMC Clinical Cohort. Participants were sampled a median of 6 days after initiation of mechanical ventilation, the median P/F ratio was 185, and 57% of the patients were diagnosed with bacterial pneumonia.

We developed an alveolar macrophage gating strategy based on the relative cell-surface protein expression levels of CD71 and CD163 that we observed with our CITE-seq data (Fig. 5A). Using this gating strategy, we identified distinct alveolar macrophage subset populations within the broader CD206+ alveolar macrophage population. There was no difference in the relative proportions of Mature Macrophages (CD71HICD163HI) between participants who remained alive and those who died (Fig. 5B). However, the proportion of CD163/LGMN Macrophages (CD71LOCD163HI) was significantly higher in participants who died vs. remained alive (Fig. 5B). We next tested whether the proportion of CD163/LGMN Macrophages was associated with decreased ventilator-free days, given that this subset has recently been associated with a "pro-fibrotic" phenotype in patients with severe

COVID-19 or idiopathic pulmonary fibrosis[10,32]. We observed a non-statistically significant trend between a higher proportion of CD163/LGMN Macrophages and fewer ventilator-free days (Fig. 5B). The proportion of CD163/LGMN Macrophages was highly correlated with soluble CD163 (sCD163) levels in BAL fluid (Fig. 5B). These analyses translate our CITE-seq data into a broader clinical context by showing that alveolar macrophage transcriptional subsets such as CD163/LGMN and Mature Macrophages can be identified using cell-surface protein markers, and that these subsets are associated with clinical outcomes such as hospital mortality.

### The proportions of alveolar myeloid subsets evolve during AHRF

Although the proportions of different alveolar monocyte and macrophage subsets have been shown to rapidly change and be associated with severity of acute lung injury in animal models[6,28,33–37], little is known about the temporal dynamics of alveolar monocyte

and macrophage subsets in humans with AHRF. We compared the transcriptional programs of alveolar monocyte/macrophage subsets in B1 vs. B2 to better understand the evolution of subsets over time in humans with AHRF. The most upregulated genes in B1 compared with B2 samples in bulk analysis (encompassing alveolar monocytes, macrophages, and DCs) were inflammatory cytokines and chemokines, such as *S100A8*, *S100A9*, and *CCL2* (Fig. 6A). These findings suggest there was a global decrease in alveolar myeloid cell inflammatory programs from enrollment to 4 days later. However, when we analyzed subsets individually, we found that the proportion of Intermediate Monocyte-Macrophages (BAL Cluster 2) and Mature Macrophages (BAL Cluster 3) significantly increased from B1 to B2, whereas the proportion of Inflammatory Monocytes (BAL Cluster 4) trended downward from B1 to B2 (Fig. 6B). These findings confirm there is significant heterogeneity in the evolution of different alveolar myeloid subsets over time that bulk transcriptional analysis does not completely capture.

We integrated our dataset with a publicly-available scRNA-seq dataset that analyzed alveolar macrophages collected from healthy human participants (HP)[8] to assess how alveolar macrophage gene expression might change over an extended period of time after AHRF. We observed that alveolar macrophages from B2 had a transcriptional signature that straddled B1 and HP, suggesting these cells may be in a transitional state of repair from acute injury to health (Fig. 6C). Known mature alveolar macrophage marker genes such as *PPARG*, *MARCO*, and *ALDH2* had highest expression in HP (Fig. 6D). On the other hand, proinflammatory chemokine genes such as *CCL2* (MCP-1) and *CXCL8* (IL-8) had highest expression in B1 with almost no expression in HP. Interestingly, some genes such as *APOE* and *TREM2* had higher expression at B2 compared with B1 or HP. *APOE* and *TREM2* have been shown to mark a macrophage phenotype that is induced by phagocytosis of apoptotic cells[38–40]. Overall, we have determined that the alveolar composition of monocyte and macrophage subsets rapidly changes over a short time period after AHRF onset.

**Table 2 | Participant Characteristics of the HMC Clinical Cohort**

| Participant Characteristics | HMC Clinical Cohort (*n* = 51) |
|---|---|
| Age | 49, 34–63 |
| % Man | 78% |
| % Trauma ICU | 75% |
| % Neuro ICU | 25% |
| Intubation-to-BAL Interval (days) | 6, 4–10 |
| P/F | 185, 164–250 |
| OI | 19, 13–27 |
| % Pneumonia | 57% |
| % Hospital Mortality | 16% |

*ICU* Intensive care unit, *P/F ratio* $P_aO_2/F_iO_2$ ratio, *OI* Oxygenation index

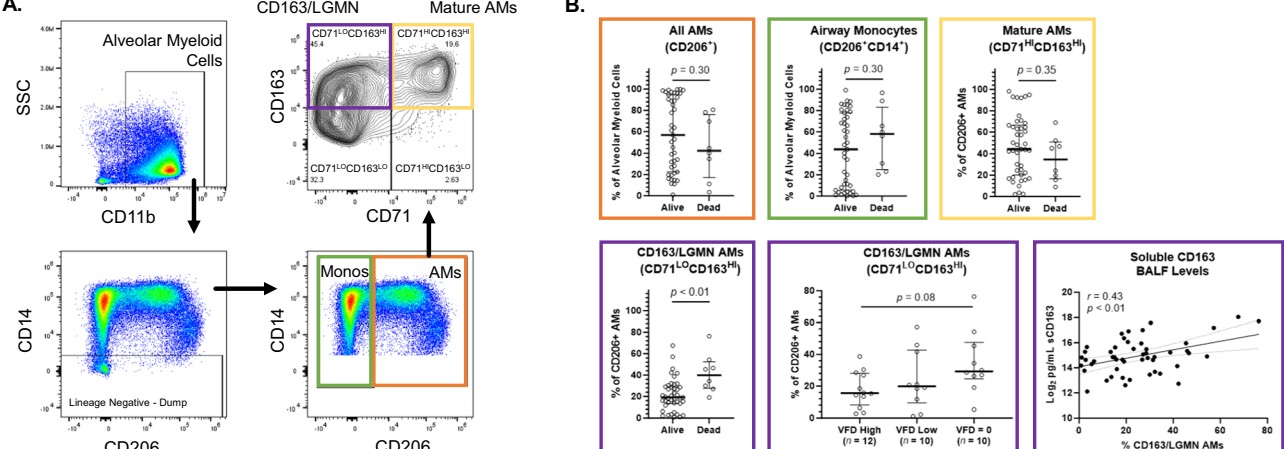

**Fig. 5 | CD163/LGMN Macrophages are Associated with Mortality in Acute Respiratory Failure.** We collected bronchoalveolar lavage (BAL) fluid from intubated and mechanically ventilated participants (HMC Clinical Cohort) (*n* = 51). Table 2 shows the participant characteristics. We analyzed alveolar cells from the BAL fluid using flow cytometry. **A** Representative gating for identifying alveolar monocytes (Monos) (green box) and CD206+ alveolar macrophages (AMs) (orange box). We classified AMs into CD71HICD163HI (yellow box – Mature AMs) or CD71LOCD163HI (purple box – CD163/LGMN AMs) subsets based on our CITE-seq data (Fig. 4C). **B** The percentage of CD206+ AMs (orange box), airway monocytes (CD206+CD14+), CD71HICD163HI (yellow box – Mature AMs), and CD71LOCD163HI (purple box – CD163/LGMN) as a proportion of all alveolar myeloid cells between participants based on hospital mortality. Depicted are the individual values, median, and interquartile range of each subset as a proportion of all alveolar myeloid

cells. *P*-values were generated with two-sided Mann-Whitney tests. CD71LOCD163HI (purple box – CD163/LGMN AMs) as a proportion of all alveolar myeloid cells between participants based on ventilator-free days (VFDs). Participants intubated > 7 days prior to bronchoscopy were excluded from this analysis. VFDs were defined as the number of days alive and free of invasive mechanical ventilation in the 21 days following bronchoscopy. VFDs were binned into tertiles. *P*-value was generated with Kruskal-Wallis test. Association between soluble CD163 BAL levels and the percentage of CD71LOCD163HI (CD163/LGMN AMs) as a proportion of all alveolar myeloid cells. Depicted are the individual values, linear regression line, and 95% confidence interval. *P*-values test whether the slope (β-coefficient) is significantly non-zero. r Pearson Correlation Coefficient.

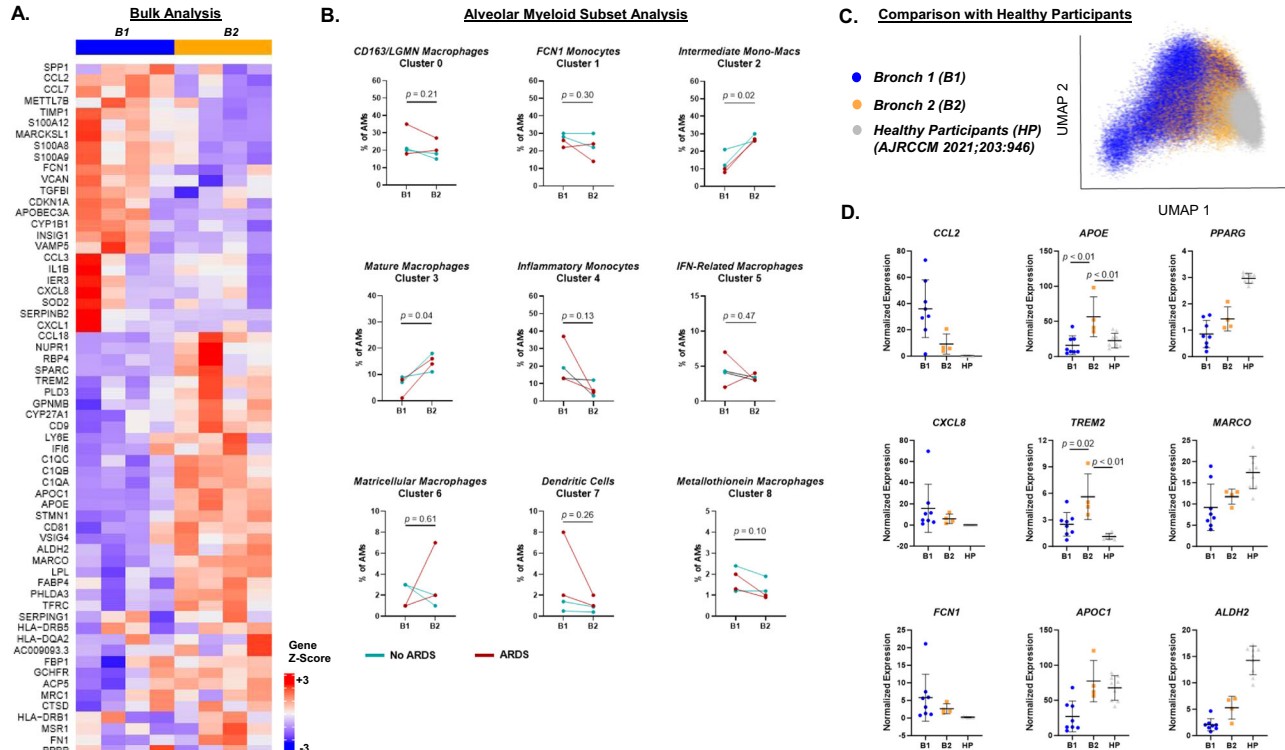

**Fig. 6 | The proportions of alveolar myeloid subsets evolve during acute hypoxemic respiratory failure. A** Heat map showing differentially expressed genes in bulk alveolar myeloid cells between Bronchoscopy 1 (B1) and Bronchoscopy 2 (B2). **B** The percentage of each alveolar myeloid subset as a proportion of all alveolar myeloid cells at B1 and B2. Depicted are the individual values and lines connecting paired samples (*n* = 4 unique participants). *P*-values were generated with paired two-sided T-tests. Participants who met criteria for ARDS are shown in red. **C** UMAP plot incorporating our dataset with single-cell gene expression data from healthy human participants (HP) who underwent BAL[8]. Color designates assignment of cells to either B1, B2, or HP. **D** Normalized gene expression of select genes between participants in B1 (*n* = 8), B2 (*n* = 4), and HP (*n* = 9). Depicted are the individual values, mean, and standard deviation of each subset as a proportion of all alveolar myeloid cells. *P*-values were generated with a two-sided T-test.

## Discussion

In this study, we leveraged CITE-seq to identify alveolar monocyte/macrophage subsets in patients with AHRF and determined cell-surface protein markers that could discriminate these subsets. We validated some of these subsets in an external clinical cohort using the higher-throughput approach of flow cytometry, and found that CD163/LGMN macrophages are associated with mortality. By analyzing paired blood and alveolar samples, we found that FCN1 alveolar monocytes, inflammatory alveolar monocytes, and IFN-related macrophages have similar transcriptional profiles as blood monocytes. In contrast, other alveolar macrophage subsets such as CD163/LGMN and Mature have highly distinct transcriptional signatures compared with peripheral blood monocytes[41]. We determined that alveolar monocytes in AHRF are uniquely classical monocytes (CD14⁺CD16⁻). Finally, we observed large changes in the relative abundance of different alveolar monocyte/macrophage subsets over the course of only 4 days during early AHRF. Collectively, our study establishes the transcriptional and phenotypic characteristics of alveolar monocytes/macrophages in early AHRF. These characteristics can be used to facilitate the identification of alveolar monocyte/macrophage subsets for future functional and clinical characterization.

The finding that CD163/LGMN macrophages represent between 20 and 40% of alveolar myeloid cells in patients with AHRF and are associated with hospital mortality strongly suggests this subset plays a key role in early AHRF (Fig. 5). In our CITE-seq analysis, CD163/LGMN macrophages were characterized by high expression of genes related to hemoglobin-haptoglobin processing (e.g., *CD163*, *HMOX1*) and MHCII antigen presentation (e.g. *LGMN*, *CTSL*) (Table S5). CD163 is a hemoglobin-haptoglobin scavenger receptor[42]. Binding of

hemoglobin-haptoglobin to CD163 on macrophages triggers anti-inflammatory responses that are mediated through heme oxygenase 1 (*HMOX1*)[43,44]. CD163 has also been shown to serve as a macrophage receptor for both gram-positive and -negative bacteria[45]. Legumin (*LGMN*) and cathepsin L (*CTSL*) are proteases that localize to the endolysosomal system. They both play a fundamental role in processing peptides to form complexes with class II MHC molecules for antigen presentation[46–49]. Reduced monocyte expression of *LGMN* has been shown to contribute to endotoxin tolerance[50]. Given their transcriptional and phenotypic characteristics, it is possible a role of CD163/LGMN macrophages in early AHRF is to efficiently process cellular and extracellular matrix debris and then direct antigen-specific adaptive immune responses. As such, we speculate the association between CD163/LGMN macrophages and mortality in our clinical cohort could be related to severity of initial lung injury and/or excessive T cell activation. Notably, CD163/LGMN macrophages have been isolated from human lung digests collected from patients with COVID-19 associated ARDS[10] as well as patients with idiopathic pulmonary fibrosis (Table S6)[32]. Although both of these studies identified CD163/LGMN macrophages within a fibrotic niche, the role of macrophages in disease is highly dependent on timing (e.g., early vs. late disease) and context (e.g. repair vs. fibrosis). Further investigation of CD163/LGMN macrophages in both time- and tissue-specific contexts is required to better understand their functional roles in AHRF and other disease states.

Our CITE-seq analysis adds clarity to prior studies that have used either transcriptional or phenotypic approaches in isolation, and confirms that mature alveolar macrophages with homeostatic transcriptional programs are best discerned by cell-surface proteins such

as CD71, CD169, and CD274 (PD-L1) (Table S9). The majority of alveolar macrophages collected from healthy humans have high expression of genes associated with homeostatic functions like surfactant processing (e.g. *FABP4*, *PPARγ*) and inhibition of inappropriate immune activation (e.g., *SERPING1*)[1,8,15,17]. Studies using cell-surface proteins to phenotype alveolar macrophages through flow cytometry or cytometry time-of-flight have determined that most alveolar macrophages from healthy individuals express CD71 (transferrin receptor), CD169 (sialoadhesin – cell adhesion), and CD274 (checkpoint protein – T cell inhibition)[3–5,51]. We and others have identified Mature Macrophages using these cell-surface protein markers in participants with ARDS[5,7] and idiopathic pulmonary fibrosis[51], however the proportions of Mature Macrophages in these disease states are much lower than in healthy participants where Mature Macrophages constitute ~90% of alveolar myeloid cells[3,5,51]. The robust finding that loss of Mature Macrophages (CD71, CD169, CD274) are associated with highly distinct pathologic states suggests the replacement of Mature Macrophages with distinct monocyte/macrophage subsets is a common feature across many forms of lung injury.

Our identification of cell-surface proteins that distinguish alveolar monocyte/macrophage transcriptional subsets provides a foundation for future studies that seek to characterize alveolar monocyte/macrophage subsets collected from large clinical cohorts. Both the CD163/LGMN and Mature subsets have high cell-surface expression of the classic alveolar macrophage marker CD206[4], making this an ineffective cell-surface marker to distinguish these two subsets (Fig. 4B). We found that a combination of CD71 and CD163 cell-surface protein expression discerned these two alveolar macrophage subsets within the CD206+ population (Fig. 5A). Consistent with findings from lung digests collected from patients with idiopathic pulmonary fibrosis[14,15], Matricellular Macrophages in our cohort of AHRF patients had high gene expression of *SPP1* (osteopontin), *MMP7*, *CHI3L1*, *GPNMB*, and *PLA2G7*. The Matricellular Macrophage subset was marked by CD163^LO^CD86^HI^ cell-surface protein expression, distinguishing it from other alveolar macrophage subsets. Future work should focus on validating and refining the cell-surface proteins that best distinguish alveolar monocyte and macrophage subsets.

Our single-cell analysis of serially-collected alveolar samples expands the existing conceptual model of how alveolar monocyte/macrophage subsets evolve over time and might contribute to injury and repair in AHRF/ARDS[6,28,33–37]. We found that the proportion of specific subsets such as Intermediate Monocyte-Macrophages and Mature Macrophages increase over early AHRF, while other populations such as Inflammatory Monocytes decrease over this same time course (Fig. 6B). This finding strongly suggests that previous associations identified between early (Day 1 of ARDS) or late (Day 8) pro-inflammatory bulk alveolar myeloid gene expression signatures and clinical outcomes are driven primarily by Inflammatory or FCN1 monocyte subsets (rather than more mature alveolar macrophage subsets)[41,52]. This finding also confirms that prior studies identifying associations between the number and proportion of alveolar monocytes/macrophages over the course of ARDS and clinical outcomes should be interpreted with caution, because different subsets within the broader alveolar macrophage population have different trajectories during AHRF/ARDS[7,53]. Future studies should consider the temporal stage of illness when analyzing alveolar monocyte/macrophage subsets given the extent their composition can change over the course of only a few days.

Our study has several limitations. First, our sample size for the CITE-seq experiments was relatively small and it is possible we did not capture all the relevant alveolar monocyte/macrophage subsets that exist in participants with AHRF. Despite this small sample size, we were able to identify both previously reported as well as potentially undescribed alveolar monocyte/macrophage transcriptional subsets. Our findings examining the CD163/LGMN subset in the larger HMC Clinical Cohort also support the validity of our CITE-seq findings. Second, we were limited to sampling participants at two timepoints separated by only 4 days. We leveraged an external dataset to infer more long-term changes in alveolar macrophage transcriptional profiles, however these findings need to be validated. Third, we did not examine the relationship between temporal changes in alveolar monocyte/macrophage subsets and clinical outcomes given our sample size. Fourth, we used RNA velocity trajectory analysis to infer cell ontogeny, however lineage tracing experiments are required to definitively demonstrate cell ontogeny. Finally, we were only able to validate CD14, CD71, and CD163 in our clinical cohort because our flow cytometry data was generated prior to our CITE-seq experiments. The "spectrum" model of macrophage activation has an inherent "line-drawing" problem that makes it challenging to classify macrophages into functionally or clinically distinct subsets[54–56]. Future studies using functional assays and more granular phenotyping approaches such as flow cytometry are required to validate the biologic and clinical relevance of the subsets we identified with CITE-seq.

In conclusion, we identify a heterogenous mixture of alveolar monocyte/macrophage subsets in patients with AHRF/ARDS, some of which are associated with hospital mortality. We provide a description of cell-surface proteins to discern alveolar monocyte/macrophage transcriptional subsets in a clinical cohort. The composition of these alveolar monocyte/macrophage subsets is highly dynamic in AHRF, supporting the general concept that they may be a modifiable factor that could be targeted to influence clinical outcomes[27,57,58].

# Methods

## Study population

All study protocols were approved and monitored by the University of Washington Human Subjects Division. We enrolled two separate cohorts for this study. The CITE-seq data was drawn from a prospectively enrolled cohort of critically ill adult patients with AHRF between 9/1/2020 and 11/8/2021 (CITE-seq Cohort). We performed research bronchoscopies and collected BAL and paired PBMCs from each participant. Written informed consent was obtained for all participants in the CITE-seq Cohort prior to any study interventions. We analyzed excess BAL fluid not needed for clinical care from a second cohort of critically ill patients undergoing bronchoscopy for suspicion of ventilator-associated pneumonia for flow cytometry experiments (HMC Clinical Cohort)[5,31]. All bronchoscopies performed in the HMC Clinical Cohort were conducted for a clinical purpose by a member of the team caring for the patient. We analyzed all extant samples with > 150,000 leukocytes that had been collected between 1/1/2016 and 8/1/21 with spectral flow cytometry. Participants in the HMC Clinical Cohort were enrolled under a waiver of informed consent. No participants in the CITE-seq or HMC Clinical Cohort were co-enrolled. Our study protocol did not include participant financial compensation.

## Research bronchoscopy

We performed the research BAL by passing a disposable fiberoptic bronchoscope (GlideScope B-Flex 5.0, Verathon product code: 0570-0375) through the endotracheal tube. Patients were preoxygenated with $F_iO_2 = 100\%$ for 5 min before and during the procedure. The procedure was performed with patients remaining on their baseline sedation and/or analgesia. We administered 1% topical lidocaine before and during the procedure as needed for cough (maximum dose = 60 mg). After the bronchoscope was wedged in the right middle lobe or lingula, five separate 30 mL aliquots of normal saline were instilled and recovered by wall-suction. An external Study Monitor reviewed all research bronchoscopy encounters and was responsible for reporting any adverse events to the University of Washington Human Subjects Division. There were no serious adverse events associated with this study.

## Clinical definitions

We abstracted clinical data from the electronic medical record into standardized case report forms. ARDS was defined by the 2012 Berlin definition[23]. Sequential organ failure assessment (SOFA) severity scores were calculated based on the original instrument[24]. Oxygenation index (OI) was calculated by dividing the product of $F_iO_2$ and mean airway pressure by the $P_aO_2$[59]. In the HMC Clinical Cohort analysis, ventilator-free days were calculated as the number of days a participants was alive and free of invasive mechanical ventilation within the 21 day period after the participant underwent bronchoscopy[60]. We excluded patients from this analysis who underwent the bronchoscopy >7 days after initiation of mechanical ventilation.

## Sample collection and processing

The BAL fluid collected for research bronchoscopy (CITE-seq analysis) or as part of the HMC Clinical Cohort (flow cytometry analysis) was processed using the same protocol. We filtered fresh BAL fluid through a 70-micron cell strainer (Fisher Scientific, Catalogue number: 08-771-2). The filtrate was centrifuged at 400 x g x 5 min and then the supernatant was separated and aliquoted. The cell-pellet was resuspended and incubated in Red Blood Cell Lysis Buffer (BioLegend, Catalogue number: 420301) x 15 min at room temperature. After incubation, the cell suspension was centrifuged at 400 x g x 5 min. The supernatant was discarded, and the remaining cell pellet was resuspended in 2% FBS/PBS. We obtained a cell count, and then aliquoted the cells at a concentration of 10,000,000 cells/mL freezing media (7% DMSO/FBS). The cells were placed into a Mr. Frosty freezing container at −80 °C to allow for controlled cooling for at least 24 h and then transferred to liquid nitrogen for longer-term storage.

Peripheral blood was collected into Cell Preparation Tubes (CPTs) and centrifuged at 1800 x g x 25 min at room temperature. The buffy coat was collected and then washed in PBS. The suspension was centrifuged at 400 x g x 15 min at room temperature. The remaining cell pellet was resuspended in 2% FBS/PBS, cells were counted, and then aliquoted at a concentration of 5,000,000 cells/mL freezing media (7% DMSO/FBS). The cells were placed into a Mr. Frosty freezing container at −80 °C to allow for controlled cooling for at least 24 h and then transferred to liquid nitrogen for longer-term storage.

## Biomarker measurements

We measured BALF biomarker concentrations using electro-chemiluminescent immunoassays per the manufacturer's instructions (Meso Scale Discovery) (V-Plex Proinflammatory Panel 1 (K15049D); V-Plex Chemokine Panel 1 (K15047D); V-Plex Cytokine Panel 1 (K15050D); U-Plex sPD-L1 (K151Z7K); R-Plex Calprotectin (K151AJYR); R-Plex Rage (K1514QR); R-Plex sCD163 (K151J4R)). All BALF samples underwent two freeze-thaw cycles prior to analysis.

## Flow cytometry

Cryopreserved cells were thawed in batches on the day of analysis in a 37 °C water bath x 2 min, resuspended in RPMI, and then centrifuged at 400 x g x 5 min. The cells were washed in RPMI and the solution was again centrifuged at 400 x g x 5 min. The cell pellet was resuspended in Benzonase (1:5000 dilution in RPMI). Cells were stained with an eFluor455UV fixable viability dye (Fisher Scientific, Catalogue number: 65-0868-14) x 30 min at room temperature in the dark. The cells were washed with PBS and then we stained the cells with an antibody cocktail x 30 min at room temperature in the dark. Table S10 shows the antibodies/clones for our flow cytometry antibody cocktail. After staining, cells were fixed with 4% PFA/FACS buffer x 10 min at room temperature in the dark. Stained cells were washed, resuspended in FACS buffer, and data was acquired on a 5-laser Cytek Aurora (Cytek Biosciences) spectral flow cytometer at an event rate for 2500–5000 events per second. We unmixed the spectral flow cytometry data using SpectraFlow software and analyzed the.fcs files with FlowJo version 10.8.1.

## Single-cell isolation and library preparation for CITE-seq

Cells were thawed in batches on the day of analysis in a 37 °C water bath x 2 min, resuspended in RPMI, and then centrifuged at 400 x g x 5 min. The cells were washed in RPMI and the solution was again centrifuged at 400 x g x 10 min. The cell pellet was resuspended in Benzonase (1:5000 dilution in RPMI) x 5 min at 37 °C. The cells were filtered through a 70-micron cell strainer into a FACS tube, washed with PBS, centrifuged at 400 x g x 5 min, and then resuspended in blocker (1:200 IgG human/mouse blocking solution) x 15 min at room temperature. The solution was centrifuged at 400 x g x 5 min and the cell pellet was resuspended in MojoSort Buffer (BioLegend, Catalogue number: 480017). We incubated cells with anti-human CD66b (Clone G10F5, BioLegend, Catalogue number: 305120, dilution 1:40) on ice x 15 min, washed the cells, and centrifuged the solution at 400 x g x 5 min. The cell pellet was resuspended in MojoSort Buffer containing the Streptavidin Nanobeads to bind to CD66b labeled cells on ice x 15 min. The solution was washed, centrifuged at 400 x g x 5 min, resuspended in MojoSort Buffer, and added to the magnet sorter for a total of three separate cycles each 5 min in duration. The cell suspension was centrifuged at 400 x g x 5 min, resuspended in PBS, and the remaining cells were counted.

The cells were first stained with Calcein AM (1:10,000, BD Pharmigen, Catalogue number: 564061) x 30 min at room temperature. The solution was washed with PBS then centrifuged at 400 x g x 5 min. The cell pellet was then stained with a MasterMix of CD15 (Clone W6D3, Biolegend, Catalogue number: 323028, dilution 1:16), CD45 (Clone H130, BioLegend, Catalogue number: 304008, dilution 1:50), Human IgG, Mouse IgG, TotalSeq-C Human Universal Cocktail v1.0 (BioLegend, Catalogue number: 399905), TotalSeq-C spike-in antibodies CD206 (TotalSeq-C0205 anti-human CD206 Clone 15-2, BioLegend, Catalogue number: 321147), and CD279 (TotalSeq-C0088 anti-human CD279 (PD-1) Clone EH12.2H7, BioLegend, Catalogue number: 329963). Table S3 shows the entire CITE-seq antibody panel. The cells were incubated in this MasterMix at 4 °C x 45 min in the dark. The solution was washed with FACS buffer, centrifuged at 400 x g x 5 min, and resuspended in FACS buffer for cell sorting. We sorted cells using an Aria Fusion and gated on live, single, $CD45^+$, $CD15^-$ cells targeting >100,000 events. The cells were then processed for single cell capture, library preparation, and sequencing.

Single cells were processed using the Chromium Next GEM Single Cell 5' v2 Kit (10x Genomics). Cells were loaded onto each channel with a target recovery of 10,000 cells. Libraries were sequenced on a NextSeq2000 (Illumina) with an average sequencing depth of 20,000 reads/cell for gene expression and 5000 reads/cell for cell-surface proteins.

## Computational analysis

We used Cell-Ranger mkfastq (10x Genomics) to demultiplex and produce raw fastq files for downstream analyses. We used Cell-Ranger multi to align per-sample reads of gene expression to the GRCh38 human reference genome, as well as feature barcoded TotalSeq antibodies to their reference sequences. We combined per-sample-count matrices into a filtered aggregated matrix using Cell-Ranger aggr. We excluded cells with < 500 genes, >4500 features, >12.5% reads mapping to the mitochondrial genome, >15% genes mapping to ribosomal genes, and >5% genes aligning to heme-associated genes.

We used query-reference mapping in Seurat to annotate cells based on their similarity to expression in the provided peripheral blood monocyte dataset[22]. We filtered our Seurat object to only include monocytes, macrophages, and classic dendritic cells as defined by this reference. To account for batch effects, we normalized each sample (method = Center Log Fold Ratio) and filtered based on the above quality control metrics independently before integrating

samples via Seurat's FindIntegrationAnchors function (number of variable anchors = 2000).

## Statistical analyses

We performed PCA analysis in Seurat[22] using both surface antibodies and gene expression data (30 PCs from gene expression and 15 PCs from surface antibodies), followed by dimensional reduction (UMAP, dims: 1:30). We calculated nearest neighbors and performed Louvain clustering in Seurat (resolution = 0.3). We identified both top gene and antibody markers for each of our clusters using FindMarkers (Wilcox-T test, $\log_2$fc > 0.25, $p$ < 0.05 for gene expression, $p$ < 0.05 and $\log_2$fc > 0.1 for antibodies). In a sensitivity analysis, we performed noise reduction of the antibody markers by utilizing three different mouse isotype negative controls to account for background technical noise generated from factors such as ambient, unbound antibody encapsulated in the droplets. We projected the cell-surface protein "antigen specificity score," which is the likelihood of an antigen binding to a specific receptor compared to the negative control ((1−beta.cdf (0.925, Antigen UMI + 1, Control UMI + 3)) ∗ 100), onto the alveolar myeloid cluster UMAP.

For our integrated blood-lung analysis, we used integration anchors in Seurat to perform query-reference mapping between our BAL dataset and paired PBMC samples. This allowed PBMC samples to be visualized on the previously produced UMAP projection of BAL cells. We annotated our PBMC samples based on cluster identities derived from our BAL dataset (Multimodal Reference Mapping), as well as calculated additional PCs ($n$ = 30) and UMAP (dims: 1:30) to describe variation in the combined PBMC and BAL dataset. We used ggplot2 to visualize our analyses[61]. We used T-tests or Mann-Whitney tests based on whether the data had a parametric or non-parametric distribution for univariate comparisons of expression levels or percent populations. Sample size for power analysis was not predetermined for our study.

We derived average gene expression values for B1 and B2 alveolar myeloid cells via the AverageExpression Function in Seurat, creating pseudo-bulked samples for further differential expression analysis. We then identified differentially expressed genes between B1 and B2 timepoints ($\log_2$fc > 0.25, $p$-value < 0.05). In order to compare our data with previously published data from healthy participants[8], we batch corrected and integrated both datasets via shared variable genes using FindIntegrationAnchors (n anchors = 2000). We averaged the scaled expression of each gene across our samples and our clusters using the pseudo-bulk approach described above.

We performed RNA velocity and trajectory analyses on the BAL samples by extracting the metadata from our filtered Seurat object and used it to construct an anndata object for processing in python using Scampy[62], NumPy[63], and pandas[64]. Loom files containing spliced and unspliced reads on a per sample basis were constructed using velocyto[29], and integrated with our pre-existing metadata with the merge utility accompanying the package scVelo[65]. We used scVelo to calculate velocity scores for each cell in our dataset, and visualized our previously constructed projections with the included velocity vectors. We generated a PAGA velocity graph using the RNA velocity data[30]. All statistical analyses were performed in R version 4.2.1 and in Python2.

## Reporting summary

Further information on research design is available in the Nature Portfolio Reporting Summary linked to this article.

## Data availability

We accessed published RNAseq data from healthy participants[8] from the GEO database under accession code GSE151928. We accessed a reference dataset[22] in Seurat to annotate our CITE-seq data using GSE164378. We accessed the GRCh38 human reference genome using assembly accession GCA_000001405.29. We have deposited the CITE-seq data generated in this study into the GEO database under accession code GSE234918. All other data are available in the article and its Supplementary files or from the corresponding author upon request. Source data are provided with this paper.

## Code availability

We have posted the code we used for the CITE-seq analysis into GitHub (https://github.com/BenaroyaResearch/Alveolar_Macrophage_Subsets_ARDS).

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

## Acknowledgements

The authors thank the research participants, their families, and the staff at HMC for their generous participation in our study. We would also like to thank Dr. Timothy Eoin West for providing oversight as the external Study Monitor and Dr. Adam Wojno as Director of the Flow Cytometry Core at BRI. Finally, we thank the following funding sources that supported this study: National Institutes of Health grant NHLBI K23HL144916 (E.D.M.) Francis Family Foundation/Parker B. Francis Fellowship (E.D.M.) National Institutes of Health grant NHLBI RO1HL149676 (C.M.) National Institutes of Health grant NIH T32HL007287-42 (S.E.H.) National Institutes of Health grant U19AI142733 (C.M., S.Z.).

## Author contributions

Authorship follows the inclusion and ethics in global research policies that are recommended by the Nature Portfolio journals. E.D.M., S.E.H., and C.M. contributed to study concept and design; E.D.M., S.E.H., M.L., M.O., Z.F., M.M., H.D., V.H.G., A.G., E.B., T.L., I.D.P., A.R., S.Z., M.M.W., C.M. contributed to protocol development and the acquisition, analysis, and interpretation of the data; M.L. and H.D. contributed to computational analysis; E.D.M., S.E.H., and C.M. contributed to literature search and initial drafting of the manuscript; E.D.M., S.E.H., M.L., M.O., Z.F., M.M., H.D., V.H.G., A.G., E.B., T.L., I.D.P., A.R., S.Z., M.M.W., C.M. contributed to and approved the final version of the manuscript; E.D.M., S.E.H., M.L., M.O., Z.F., M.M., H.D., V.H.G., A.G., E.B., T.L., I.D.P., A.R., S.Z., M.M.W., C.M. agree to be accountable for all aspects of the work.

## Competing interests

The authors declare no competing interests.
