## [Peer Review File · Nature Communications]

The Transcriptional and Phenotypic Characteristics that Define Alveolar Macrophage Subsets in Acute Hypoxemic Respiratory FailureREVIEWER COMMENTS

Reviewer #1 (Remarks to the Author):

The authors utilized CITE-seq analysis (scRNA-seq coupled to surface protein quantification) for a characterization of alveolar monocyte and macrophage populations. Next to comparing different sub populations to known subtypes, they also identify novel populations and determine a set of surface markers that can be used for further testing. Improving our understanding of immune cell populations in health and disease is very relevant and with CITE-seq the authors have chosen a very suitable and novel technology that can provide more insights into immune characterization.

While overall design and methods are sound, I have a few comments and concerns.

1) As one of their main findings the authors determine a parsimonious set of surface markers (CD14, CD163, CD123, CD71, CD48, CD86, and CD44) to be used in future clinical studies. However, in their own validation study using flow-cytometry on a second clinical cohort, four out of those seven markers are not included despite they were described as necessary for the distinction of particular subpopulations (CD123, CD48, CD86, and CD44). This makes it difficult to assess their actual relevance.

Why were those four markers excluded from the validation?

2) The authors provide particular QC thresholds for excluding cells (< 500 genes, > 4,500 genes, or had > 12.5% of reads mapping to mitochondrial genes). How were those thresholds defined? Importantly, why is the threshold for reads mapping to mitochondrial genes so low? Especially during immune response the mitochondrial content in immune cells can rise to meet increased energy requirements. Thus, many cells (particular activated ones) might have been excluded based on this low threshold. E.g. a threshold of 30% might be more adequate to not risk losing interesting cells.

3) When investigating surface markers a common problem is the noise and unspecific binding of antibodies. Commonly in CITE-seq analyses, truly positive cells have to be distinguished from spuriously labelled ones. Also in Figure 3 it becomes apparent that for

most markers the majority of cells seems to be positive, despite relative differences between the clusters. Can the authors please comment on whether noise reduction would improve the analysis, potentially also bringing out more differences between the monocyte and macrophage subsets? Moreover, it would be helpful to visualize the surface marker expression using UMAP and/or a dot plot.

Minor:

a) In Figure 1D it would be helpful to combine the plots for Bronchoscopy 1 and 2 into one plot (showing the %cells for B1 and B2 always next to each other), to allow an easier and more direct comparison between the time points.

b) Figure 2D lists “pseudogenes” for cluster 6. Where does this label come from? Isn't cluster 6 the matricellular AMs?

c) The B1 versus B2 comparison that is shown in Figure 5 is not properly described in the methods. Figure 5A claims a bulk analysis, but in the methods it is only written (line 501) “We calculated additional PCAs and a UMAP using only genes significant between B1 and B2 timepoints ($\log_2fc > 0.25$, $p\text{-value} < 0.05$) for our time-series analysis”. How was the differential gene expression done when this was a bulk analysis? Please describe which methods were used.

d) The CITE-seq data has not yet been deposited, but should be uploaded before manuscript acceptance.

Reviewer #2 (Remarks to the Author):

Comments to the Authors

General Comments: In this manuscript by Morrell and colleagues, the authors employ CITE-

seq (single-cell RNA-sequencing and cell surface protein quantification) to characterize macrophage/monocyte heterogeneity in matched PMBC and bronchoalveolar lavage cells collected in a small cohort of patients with acute hypoxic respiratory failure. These experiments and analyses found 2 monocyte and 6 macrophage subsets in alveolar samples using this technology. Importantly, they compare their findings to work from prior scRNA-seq studies by other groups. The authors found that some subsets identified in this study overlapped with macrophage subsets found in healthy subjects or patients with COVID-19 from these other studies. However, several new subsets were identified by the transcriptional signatures in this study. Moreover, the time points evaluated demonstrate how rapidly these subsets change in the injured and resolving lung.

The CITE-seq method allowed for the opportunity to determine potential surface markers which may be utilized to distinguish alveolar macrophage subsets in future studies. The authors validated this approach in a separate external cohort dataset.

This study is well performed, clear, concise, and adds to the field. The methods are detailed and the work supports the authors' conclusions. There are several minor comments.

Minor Comments:

1. Prior work from this group has shown an association with CD274 (PDL1) and/or CD169 expression on macrophages from different clinical populations (ref 5). These markers have the highest expression in cluster 3 from the cell surface data set presented in this study. While the methods and clinical populations are different between these studies, I wonder if there is an interpretation to be made between this group's prior work and this study. This may not be the case, and maybe these markers did not fall out in this work due to the enhanced granularity provided by CITE-seq and the analyzes employed.

2. Most participants' primary risk factor was trauma (6/8), with 75% of the participants with ARDS in this cohort having trauma as the primary risk factor. Two of the participants received massive transfusion protocols. Did other trauma patients receive PRBCs? Given potential phenotypic expression differences of surface markers or gene expression between donor and recipient, would this affect findings (in the blood, maybe the BAL)? Additionally, would this be a way to delineate recruited donor cells (from participants receiving PRBCs) in

the lavagable compartment, such as with HLA typing, which could be determined from 5' scRNA-seq samples?

3. On page 17, lines 408 and 409, it reads that the supernatant fluid after the RBC lysis is aliquoted and stored for downstream analysis. Was the RBC lysis supernatant what was tested for in this study, or was the BAL supernatant fluid after the filtering and the first centrifugation for the cell pellet what was tested?

Reviewer #3 (Remarks to the Author):

In the context of this study, the authors had access to two cohorts of human patients: a first cohort of 12 patients hospitalised for Acute Hypoxemic Respiratory Failure (of various etiologies) from which they isolated cells from bronchoalveolar lavage and blood leukocytes to perform an elegant CITE-seq analysis (single-cell RNA-sequencing + cell-surface protein quantification). They then sought to validate the functional relevance of certain surface markers (including the CD163 and CD71 molecules) by performing multiparametric flow cytometry analyses on 51 patients hospitalised (HMC Clinical Cohort) for severe pneumonia requiring ventilation. Despite the interest of the questions addressed on human samples, and the cutting-edge technologies used, this study remains too descriptive and too superficial to be published in a journal such as Nature Communications. There is a lack of functional investigations to highlight the biological relevance of the observations made. Many assumptions are made based on correlations, but there is a lack of scientific evidence to support the conclusions proposed in this paper.

Major comments

- This is a descriptive study based on interesting human samples, but it lacks functional investigations and a clear association with clinical outcome. Based on the results generated, the authors make various extrapolations that are not confirmed by functional mechanistic investigations. Part of the discussion consists of speculation, which may over-interpret the results. For example, the statement "Our findings support an overall model of AHRF/ARDS pathogenesis in which highly diverse alveolar monocyte/macrophage populations mediate all aspects of the host response to injury – including repair – almost immediately after an

inciting event". In what way do these data recapitulate the overall model of AHRF/ARDS pathogenesis? More importantly, what makes it possible to say that certain subsets, defined based on the sc-RNAseq analysis, are the cells that mediate the deleterious/beneficial effects? How can we distinguish between causes, consequences and innocent bystander effects? There is a lack of evidence supporting the interpretation and the conclusions.

- While it is extremely interesting to have access to human samples and therefore to start from a situation encountered in humans, this study will benefit from parallel investigations carried out in mice. It would be interesting to have an integrated analysis of single cell data in humans and mice. This would greatly help to validate certain correlations and extrapolations that have been made, and would make it possible to better address the question of causal effect versus consequence or non-relevant association.

- The definition of the different clusters is not very clearly explained or justified by references in the literature. It is conventional to refer to tissue resident macrophages and monocyte-derived alveolar macrophages. What is the basis for the definition of the different clusters?

- It is somewhat regrettable that the scRNA-seq was analysed by pooling the data from the various patients (or at least presenting it as a 'data pool'). It would be more interesting to tag the samples individually (sample tag) and make correlations between severity/etiology/age and the definition of a particular transcriptomic profile. Indeed, among others, one parameter that must also be taken into account when studying the heterogeneity of alveolar and/or pulmonary macrophage populations is ageing. This has not been discussed at all, despite the fact that the patients enrolled in the cohort are of different ages. It would therefore be interesting to stratify patients by severity and also by age.

- The authors claim to have found a combination of surface markers enabling them to discriminate between the different subsets identified, but these subsets are not associated with any particular functional properties in the first dataset, nor with a clear clinical outcome/prognosis, which limits the scope of these results. Interestingly, the authors

wanted to investigate the functional relevance of the selected markers in a second cohort of human patients hospitalised for severe pneumonia requiring ventilation. Despite the development of a large panel (table S8), only the markers CD163 and CD71 were considered and shown in the analysis, which is unfortunate given that the CD71- alveolar macrophage subset has already been described and associated with a poor prognosis in various clinical entities. (IPF : Am J Respir Crit Care Med. 2019, PubMed: 31051082 ; COVID-19 : Cell 184, 6243-6261.e27 (2021)).

- Did the authors identify CD71 in their scRNA-seq analysis? It would be interesting to show features plot UMAPs with CD163 and CD71 expression.

Minor comments:

- Some abbreviations have not been introduced (e.g. LGMN, FCN1). The writing style could be reworked.

- There are some 'oversimplifications'. For example, overexpression of the CD86 marker is associated with the production of Th2 cytokines (discussion p 12), whereas many other studies associate this marker with 'M1' type polarization of alveolar macrophages (e.g. Plos one DOI: 10.1371/journal.pone.0045466; Nat. Immunol. DOI: 10.1038/ni.1937 ; Curr Opin Immunol DOI: 10.1016/j.coi.2010.01.009).

- The discussion should be balanced and enriched by more references and comparisons with data in the literature.

- Analysis of developmental trajectory provides an indication, a bioinformatic assessment to infer the fate of a specific cell type but not a definitive demonstration of cell ontogeny.

- It would be interesting to discuss the role of CD71 and CD163 molecules.

- The authors stated that it is not known whether alveolar monocyte and macrophage subsets that have been identified by scRNA-seq in subjects with COVID-19 are conserved in critically-ill patients without viral infection. Therefore, it would be interesting to compare

the observations made in the present study with the results published for patients with COVID-19.

Reviewer 1 General Comments: *The authors utilized CITE-seq analysis (scRNA-seq coupled to surface protein quantification) for a characterization of alveolar monocyte and macrophage populations. Next to comparing different sub populations to known subtypes, they also identify novel populations and determine a set of surface markers that can be used for further testing. Improving our understanding of immune cell populations in health and disease is very relevant and with CITE-seq the authors have chosen a very suitable and novel technology that can provide more insights into immune characterization.*

While overall design and methods are sound, I have a few comments and concerns.

Response: Thank you very much for taking the time to extensively review our manuscript and for providing your thoughtful comments. We address each of your specific comments below.

Reviewer Comment #1: *As one of their main findings the authors determine a parsimonious set of surface markers (CD14, CD163, CD123, CD71, CD48, CD86, and CD44) to be used in future clinical studies. However, in their own validation study using flow-cytometry on a second clinical cohort, four out of those seven markers are not included... This makes it difficult to assess their actual relevance. Why were those four markers excluded from the validation?*

Response to Comment #1: The reason we did not include all the cell-surface proteins we identified with CITE-seq in our flow cytometry analysis (validation cohort) is that the flow cytometry data was generated prior to our CITE-seq findings. We independently hypothesized CD71, CD163, and CD14 were clinically relevant prior to our CITE-seq experiments, which is why they were included the flow cytometry panel. Unfortunately, there are not sufficient samples remaining from these subjects to perform additional analyses. Alveolar leukocytes from critically ill subjects with acute respiratory failure are extremely difficult to obtain. It is not feasible to develop a similar cohort and we are unaware of any existing datasets from other groups we could leverage to validate the findings.

In order to better assess the clinical and biologic relevance of the subset identified by CD123, CD48, CD86, and CD44, we have included new analyses testing for associations between these subsets and clinical/biomarker parameters. This was suggested by Reviewer 3. Although these analyses are performed on a small sample set, we identified associations between the subsets and clinical and biomarker profiles (Figure 2, shown below). We think the inclusion of these additional analyses strengthens the argument for the relevance of these subsets, although we acknowledge in the revised discussion section the need for further validation (Discussion Section - paragraph #6).

Figure 2. Correlations Between Alveolar Myeloid Subsets, Biomarker Profiles, and Clinical Severity

Figure 2. Correlations Between Alveolar Myeloid Subsets, Biomarker Profiles, and Clinical Severity. (A) Heatmap of the correlation coefficients between alveolar myeloid subset proportions (y-axis) and log₂ alveolar biomarker levels (x-axis). Colors represent the correlation with scale indicating value of Pearson's r correlation. Axes are ordered by clustering based on Pearson correlation-distances using pheatmap. (B) Associations between the proportion of alveolar myeloid subsets as a percentage of all alveolar myeloid cells (y-axis) and oxygenation index (OI) (x-axis). OI is a measure of respiratory failure severity that accounts for

both oxygenation and mean airway pressure being delivered by mechanical ventilation. Higher values indicate more severe respiratory failure. Depicted are the individual values, linear regression best-fit line, and 95% confidence intervals. P-values test whether the slope (β -coefficient) is significantly non-zero. * designates P-value < 0.05. **(C)** The percentage of each alveolar myeloid subset as a proportion of all alveolar myeloid cells in subjects with or without ARDS. P-values were generated with a Mann-Whitney test.

Reviewer Comment #2: *The authors provide particular QC thresholds for excluding cells (< 500 genes, > 4,500 genes, or had > 12.5% of reads mapping to mitochondrial genes). How were those thresholds defined? Importantly, why is the threshold for reads mapping to mitochondrial genes so low? Especially during immune response the mitochondrial content in immune cells can rise to meet increased energy requirements. Thus, many cells (particular activated ones) might have been excluded based on this low threshold. E.g. a threshold of 30% might be more adequate to not risk losing interesting cells.*

Response to Comment #2: We used a QC threshold of excluding cells with > 12.5% of reads mapping to mitochondrial genes for multiple reasons. First, there is a strong right-skewed tapered distribution of cells with reads mapping to mitochondrial genes in our dataset past the 12.5% threshold (Figure S3). Importantly, there is no evidence of a bimodal distribution of cells with reads mapping to mitochondrial genes beyond the 12.5% threshold. These findings strongly suggest our QC threshold is not excluding an important population of viable cells. Second, the genes most highly expressed in the cells with > 12.5% of reads mapping to mitochondrial genes strongly suggests these are dying cells. The two most differentially expressed genes in the cells with > 12.5% of reads mapping to mitochondrial reads vs. cells with < 12.5% of reads mapping to mitochondrial genes were *NEAT2* (also known as *MALAT1*) (\log_2 FC 1.43, p-value < 1×10^{-95}) and *NEAT1* (\log_2 FC 1.33, p-value < 1×10^{-95}). *NEAT2* and *NEAT1* (noncoding nuclear-enriched abundant transcript 1 and 2) are almost exclusively expressed in the cell nucleus (*BMC Genomics* 2007;8:39) and have been consistently observed in dead/dying cells detected in poly-A captured scRNA-seq data (10X Genomics website: <https://kb.10xgenomics.com/hc/en-us/articles/360004729092-Why-do-I-see-high-levels-of-Malat1-in-my-gene-expression-data->). Finally, our QC threshold of excluding cells with > 12.5% of reads mapping to mitochondrial genes is consistent with other reports that have systematically examined the proportion of mitochondrial genes across human tissues where no cell types had average mitochondrial reads > 10% (*Bioinformatics* 2021;37(7):963). The high-end of the IQR for mitochondrial reads in monocyte-derived macrophages was below 12.5% in this analysis.

To address your very important concern, we have included new Figures S2 and S3 (shown below). These figures display the histograms of our CITE-seq QC data, which strongly suggest our thresholds did not exclude viable cell populations. We thank the reviewer for encouraging us to include these data, which we feel improves the rigor of our manuscript.

Figure S2. Distribution of the Number of Genes and Cells Detected for Each Subject

Figure S2. Distribution of the Number of Genes per Cell Detected for Each Subject. Histograms displaying the distribution of genes (x-axis) per cell (y-axis) for each sample. We excluded cells with < 500 or > 4,500 genes (designated with red line).

Figure S3. Distribution of Percentage of Reads Mapping to Mitochondrial Genes per Cell for Each Subject

Figure S3. Distribution of Percentage of Reads Mapping to Mitochondrial Genes per Cell for Each Subject. Histograms displaying the % of reads mapping to mitochondrial genes (x-axis) per cell (y-axis) for each sample. We excluded cells with > 12.5% of reads mapping to mitochondrial genes (designated with red line). The two most differentially expressed genes in the cells with > 12.5% of reads mapping to mitochondrial reads vs. cells with < 12.5% of reads mapping to mitochondrial genes were *NEAT2* (also known as *MALAT1*) (\log_2 FC 1.43, p-value < 1×10^{-95}) and *NEAT1* (\log_2 FC 1.33, p-value < 1×10^{-95}). *NEAT2* and *NEAT1* (noncoding nuclear-enriched abundant transcript 1 and 2) are almost exclusively expressed in the cell nucleus (*BMC Genomics* 2007;8:39) and have been consistently observed in dead/dying cells detected in poly-A captured scRNA-seq data.

Reviewer Comment #3: *When investigating surface markers a common problem is the noise and unspecific binding of antibodies... Can the authors please comment on whether noise reduction would improve the analysis, potentially also bringing out more differences between the monocyte and macrophage subsets? Moreover, it would be helpful to visualize the surface marker expression using UMAP and/or a dot plot.*

Response to Comment #3: Thank you for this very helpful suggestion. We performed noise reduction by utilizing three different mouse isotype negative controls to account for background technical noise generated from factors such as ambient, unbound antibody encapsulated in the droplets. We projected the cell-surface protein “antigen specificity score,” which is the likelihood of an antigen binding to a specific receptor compared to the negative control ($(1 - \text{beta.cdf}(0.925, \text{Antigen UMI} + 1, \text{Control UMI} + 3)) * 100$), onto the alveolar myeloid cluster UMAP (Figure S8, shown below). This noise reduction approach did not significantly alter our primary findings, however we agree this new figure is a helpful complement to Figure 4 (this was also recommended by Reviewer #3).

Figure S8. Cell-Surface Antigen Specificity Scores

Figure S8. Cell-Surface Protein Specificity Scores. (A) Uniform manifold approximation and projection (UMAP) displaying the alveolar myeloid cells colored by subsets. (B, C, and D) Antigen-specificity scores projected onto the alveolar myeloid cell UMAP. The antigen-specificity score is the likelihood of an antigen binding to its receptor compared with the negative isotype control. (B) Canonical alveolar macrophage (CD206), monocyte (CD14), and dendritic cell (CD1c) cell-surface protein markers. (C) Most differentially expressed cell-surface proteins across the alveolar macrophage subsets. (D) Most differentially expressed cell-surface proteins across the alveolar monocyte subsets.

Reviewer Minor Comment #1: *In Figure 1D it would be helpful to combine the plots for Bronchoscopy 1 and 2 into one plot (showing the %cells for B1 and B2 always next to each other), to allow an easier and more direct comparison between the time points.*

Response to Minor Comment #1: Thank you for this suggestion. We have modified Figure 1D so the plots for Bronchoscopy 1 and 2 are directly juxtaposed as you suggested.

Reviewer Minor Comment #2: *Figure 2D lists “pseudogenes” for cluster 6. Where does this label come from? Isn’t cluster 6 the matricellular AMs?*

Response to Minor Comment #2: Thank you for highlighting this area of our manuscript that was not clear. Cluster 6 in the original Figure 2D was generated from a separate analysis performed on pooled alveolar and blood samples. To make this clearer, we moved Figure 2D to Figure S7, which displays the data from the rest of the integrated alveolar/blood analyses.

Reviewer Minor Comment #3: *The B1 versus B2 comparison that is shown in Figure 5 is not properly described in the methods...How was the differential gene expression done when this was a bulk analysis? Please describe which methods were used.*

Response to Minor Comment #3: Thank you for highlighting this omission in our methods section. We have edited the methods section to explain how we performed the bulk RNA-seq analysis in Figure 6 (Figure 5 in the first submission).

Reviewer Minor Comment #4: *The CITE-seq data has not yet been deposited, but should be uploaded before manuscript acceptance.*

Response to Minor Comment #4: We have uploaded the CITE-seq data into Gene Expression Omnibus and added the accession number to the manuscript (GSE234918).

Reviewer 2 General Comments: *In this manuscript by Morrell and colleagues, the authors employ CITE-seq (single-cell RNA-sequencing and cell surface protein quantification) to characterize macrophage/monocyte heterogeneity in matched PMBC and bronchoalveolar lavage cells collected in a small cohort of patients with acute hypoxic respiratory failure. These experiments and analyses found 2 monocyte and 6 macrophage subsets in alveolar samples using this technology. Importantly, they compare their findings to work from prior scRNA-seq studies by other groups. The authors found that some subsets identified in this study overlapped with macrophage subsets found in healthy subjects or patients with COVID-19 from these other studies. However, several new subsets were identified by the transcriptional signatures in this study. Moreover, the time points evaluated demonstrate how rapidly these subsets change in the injured and resolving lung.*

The CITE-seq method allowed for the opportunity to determine potential surface markers which may be utilized to distinguish alveolar macrophage subsets in future studies. The authors validated this approach in a separate external cohort dataset.

This study is well performed, clear, concise, and adds to the field. The methods are detailed and the work supports the authors' conclusions. There are several minor comments.

Response: Thank you very much for taking the time to extensively review our manuscript and for providing your thoughtful comments. We address each of your specific comments below.

Reviewer Minor Comment #1: *Prior work from this group has shown an association with CD274 (PDL1) and/or CD169 expression on macrophages from different clinical populations (ref 5). These markers have the highest expression in cluster 3 from the cell surface data set presented in this study. While the methods and clinical populations are different between these studies, I wonder if there is an interpretation to be made between this group's prior work and this study. This may not be the case, and maybe these markers did not fall out in this work due to the enhanced granularity provided by CITE-seq and the analyzes employed.*

Response to Minor Comment #1: Thank you for encouraging us to better contextualize our findings with published work in the field.

We have added a new paragraph in the discussion section (paragraph #3) that specifically addresses your comment about PD-L1 and CD169. We speculate the “AM-1” population we previously identified in Reference #5 is the same population as “Cluster 3 – Mature Macrophages” in our current study. CD169 and CD274 (PD-L1) were the 2nd and 5th most discriminatory cell-surface proteins in our CITE-seq data (CD71 was the most discriminatory) out of 126 total cell-surface proteins we analyzed (Table S8). Although our clustering approach in the prior study identified a CD169^HPD-L1^{LO} subset (“AM-2”), the distribution of CD169 and PD-L1 cell-surface expression between the different subsets in that study is consistent with our findings using CITE-seq.

We also added a few sentences in the limitations section to acknowledge the “line-drawing” problem of macrophage classification and to highlight the importance of future functional studies to define the biologic/clinical relevance of subsets we identified with CITE-seq. Finally, we have added Table S5 (as recommended by Reviewer #3 Comment #3), which compares and

contrasts our study's findings with existing literature. We think these additions to the manuscript facilitate a more complete and nuanced interpretation of our findings.

Reviewer Minor Comment #2: *Most participants' primary risk factor was trauma (6/8), with 75% of the participants with ARDS in this cohort having trauma as the primary risk factor. Two of the participants received massive transfusion protocols. Did other trauma patients receive PRBCs? Given potential phenotypic expression differences of surface markers or gene expression between donor and recipient, would this affect findings (in the blood, maybe the BAL)? Additionally, would this be a way to delineate recruited donor cells (from participants receiving PRBCs) in the lavagable compartment, such as with HLA typing, which could be determined from 5' scRNA-seq samples?*

Response to Minor Comment #2: Thank you for raising the concern that the large amount of blood transfusions some of the subjects received could have influenced our leukocyte analyses. We do not think we measured a significant amount of donor WBCs in our samples because all transfused blood products at our center are leukoreduced. Our clinical blood bank uses Pall RC-2/RC-3 leukocyte reduction filters to remove leukocytes from donor blood products. This filter is certified to result in a >4 log reduction in leukocytes (e.g. a donor unit of blood goes from 10,000 cells/microliter to < 1 cell/microliter).

To address your important concern, we have included a new Table S2 which includes the total amount of blood products each subject was transfused as well as an estimate of the total number of donor WBCs that were transfused in each subject (see below). Given the extremely small number of donor WBCs that were in each subject, we do not think a donor/recipient cell-tagging approach is necessary (or possible).

ID	Primary Risk Factor	Secondary Risk Factors	# Units PRBCs Transfused	# Units WB Transfused	Estimated Total Donor WBCs Transfused	Recipient WBCs (cells/μL)	Estimated Total Recipient WBC	% Donor WBCs in Recipient
1	Trauma	Contusions/ Massive Transfusion	3	3	4.7 cells	9.80	4.9×10^7	$9.6 \times 10^{-8}\%$
2	Trauma	Contusions	5	5	7.8 cells	14.16	7.1×10^7	$1.1 \times 10^{-7}\%$
3	Trauma	Contusions	1	0	0.8 cell	7.00	3.5×10^7	$2.3 \times 10^{-8}\%$
4	Trauma	Pneumonia/ Contusions	0	0	NA	16.34	8.2×10^7	NA
5	Aspiration Pneumonitis	VT/VF Arrest	0	0	NA	7.60	3.8×10^7	NA
6	Trauma	Pneumonia/ Massive Transfusion	13	4	13.2 cells	13.01	6.5×10^7	$2.0 \times 10^{-7}\%$
7	NSTI/Sepsis	Volume Overload	0	0	NA	14.12	7.1×10^7	NA
8	Trauma	Pneumonia	0	0	NA	10.89	5.4×10^7	NA

Reviewer Minor Comment #3: *On page 17, lines 408 and 409, it reads that the supernatant fluid after the RBC lysis is aliquoted and stored for downstream analysis. Was the RBC lysis supernatant what was tested for in this study, or was the BAL supernatant fluid after the filtering and the first centrifugation for the cell pellet what was tested?*

Response to Minor Comment #3: Thank you for identifying this error in how we explained our protocol. As you point out, the BAL supernatant was removed prior to the addition of RBC lysis buffer to the cell pellet. We have corrected the methods section.

Reviewer #3 General Comments: *In the context of this study, the authors had access to two cohorts of human patients: a first cohort of 12 patients hospitalised for Acute Hypoxemic Respiratory Failure (of various etiologies) from which they isolated cells from bronchoalveolar lavage and blood leukocytes to perform an elegant CITE-seq analysis (single-cell RNA-sequencing + cell-surface protein quantification). They then sought to validate the functional relevance of certain surface markers (including the CD163 and CD71 molecules) by performing multiparametric flow cytometry analyses on 51 patients hospitalised (HMC Clinical Cohort) for severe pneumonia requiring ventilation. Despite the interest of the questions addressed on human samples, and the cutting-edge technologies used, this study remains too descriptive and too superficial to be published in a journal such as Nature Communications. There is a lack of functional investigations to highlight the biological relevance of the observations made. Many assumptions are made based on correlations, but there is a lack of scientific evidence to support the conclusions proposed in this paper.*

Response: Thank you very much for taking the time to extensively review our manuscript and for providing your thoughtful comments.

We performed new analyses testing for associations between subsets and clinical parameters such as age, oxygenation index, and presence of ARDS as you suggested. We found that some of the subsets such as inflammatory monocytes were associated with more severe lung injury. We also performed a new integrated analysis incorporating myeloid subset proportions with 30 different soluble alveolar mediator measurements, which revealed distinct cell-cytokine profiles reflective of immune responses in non-COVID-associated AHRF. Finally, we significantly expanded our discussion of the most robust aspects of our study (e.g. Mature and CD163/LGMN macrophages) while also eliminating the more speculative aspects that you highlighted. Collectively, we think these new analyses and revised discussion section better support the functional and clinical relevance of the alveolar myeloid subsets we have identified.

We address each of your specific comments below.

Reviewer Comment #1: *This is a descriptive study based on interesting human samples, but it lacks functional investigations and a clear association with clinical outcome. Based on the results generated, the authors make various extrapolations that are not confirmed by functional mechanistic investigations. Part of the discussion consists of speculation, which may over-interpret the results. For example, the statement “Our findings support an overall model of AHRF/ARDS pathogenesis in which highly diverse alveolar monocyte/macrophage populations mediate all aspects of the host response to injury – including repair – almost immediately after an inciting event”. In what way do these data recapitulate the overall model of AHRF/ARDS pathogenesis? More importantly, what makes it possible to say that certain subsets, defined based on the sc-RNAseq analysis, are the cells that mediate the deleterious/beneficial effects? How can we distinguish between causes, consequences and innocent bystander effects? There is a lack of evidence supporting the interpretation and the conclusions.*

Response to Comment #1: We have added two major analyses into the revised manuscript to better support the biologic and clinical relevance of the alveolar myeloid subsets we identified. First, as you recommended in Comment #4, we tagged the patient samples individually and tested for associations between the alveolar myeloid subsets and clinical parameters such as age and severity of lung injury (Figure 2B). We did not identify significant associations between subsets and age, although our sample sizes are small. In contrast, we found that a higher proportion of inflammatory monocytes were associated with worse oxygenation index. Second,

we made additional alveolar biomarker measurements and generated an integrated cell-cytokine analysis (Figure 2A). We identified distinct cell-cytokine immune profiles that better contextualize how the subsets interact with their alveolar microenvironment. For example, we found that higher proportions Mature and Intermediate Monocyte-Macrophages were positively correlated with IL-13, however inversely correlated with T_H1 mediators such as CXCL10 and other proinflammatory cytokines like IL-12b.

We acknowledge it is not possible to determine causality in these human studies. However, we think our analyses have identified highly relevant alveolar myeloid subsets present in humans with AHRF/ARDS that will provide a key starting point for future mechanistic studies.

Reviewer Comment #2: *While it is extremely interesting to have access to human samples and therefore to start from a situation encountered in humans, this study will benefit from parallel investigations carried out in mice. It would be interesting to have an integrated analysis of single cell data in humans and mice. This would greatly help to validate certain correlations and extrapolations that have been made, and would make it possible to better address the question of causal effect versus consequence or non-relevant association.*

Response to Comment #2: We are hoping this paper will be a guide for future functional analyses and studies in models where mechanisms can be better elucidated. Although murine models can help uncover mechanism, one of the main limitations of using them in macrophage studies of AHRF/ARDS is there are limited numbers of functional cellular and molecular homologs between humans and mice. The overlap between the top 1,000 genes for corresponding human and mouse alveolar myeloid subsets is only 13-28% (*Cell Reports* 2020;Nov 3;33(5):108337). There are no accepted models of acute lung injury in mice that recapitulate all the features of human ARDS (*AJRCMB* 2022;66(2):e1), therefore it is not possible to discern alveolar myeloid subsets in animal models that will truly reflect subsets present in the microenvironment of human AHRF. Finally, the consensus cell-surface markers to define macrophage subsets in humans and mice are different (*AJRCMB* 2019;61(2):150).

Alveolar specimens from critically-ill patients are extremely difficult to obtain, and we think our study provides unique and valuable information that fills a key existing knowledge gap in the immunology and critical care literature.

Reviewer Comment #3: *The definition of the different clusters is not very clearly explained or justified by references in the literature. It is conventional to refer to tissue resident macrophages and monocyte-derived alveolar macrophages. What is the basis for the definition of the different clusters?*

Response to Comment #3: We address this important concern in two ways. First, we have incorporated alveolar biomarker data into our analyses that adds further support for the annotations we provide. For example, the proportion of IFN-Related Macrophages were very highly correlated with BAL levels of IFN-stimulated mediators such as CXCL10. The proportion of Inflammatory Monocytes was very highly correlated with IL-6. These alveolar biomarker profiles complement the RNA-seq and cell-surface protein data and provide additional support for the definitions of the clusters we provide. Second, we have added a new Table S5 that compares our findings with findings from published datasets. This table clarifies which clusters we speculate are tissue-resident vs. monocyte-derived.

Reviewer Comment #4: *It is somewhat regrettable that the scRNA-seq was analysed by pooling the data from the various patients (or at least presenting it as a 'data pool'). It would be more interesting to tag the samples individually (sample tag) and make correlations between severity/etiology/age and the definition of a particular transcriptomic profile. Indeed, among others, one parameter that must also be taken into account when studying the heterogeneity of alveolar and/or pulmonary macrophage populations is ageing. This has not been discussed at all, despite the fact that the patients enrolled in the cohort are of different ages. It would therefore be interesting to stratify patients by severity and also by age.*

Response to Comment #4: Thank you for this important suggestion to test whether the subsets we identified with CITE-seq are related to age and/or clinical severity. We have incorporated four new analyses into our revised manuscript testing for associations between the alveolar myeloid subsets and clinical parameters (Figure 2A/B/C, Figure S4, and Figure S5).

We did not detect associations between alveolar myeloid subsets and participant age (Figure S5). Although our sample sizes are limited, these findings are consistent with other reports that have not identified unique AM transcriptional subsets that were associated with older adults (*JCI* 2021;131:140229). We did identify associations between some subsets such as Inflammatory Monocytes and worse oxygenation index (Figure 2B). Finally, we report multiple associations between alveolar myeloid subsets and biomarker levels reflective of different inflammatory and injury profiles (Figure S4). We think these additional analyses better contextualize the alveolar myeloid subsets we identified within a clinical context and strengthen the argument for their biologic and clinical relevance. We discuss these new findings in the revised results and discussion section of the manuscript.

Reviewer Comment #5: *The authors claim to have found a combination of surface markers enabling them to discriminate between the different subsets identified, but these subsets are not associated with any particular functional properties in the first dataset, nor with a clear clinical outcome/prognosis, which limits the scope of these results. Interestingly, the authors wanted to investigate the functional relevance of the selected markers in a second cohort of human patients hospitalised for severe pneumonia requiring ventilation.*

Despite the development of a large panel (table S8), only the markers CD163 and CD71 were considered and shown in the analysis, which is unfortunate given that the CD71- alveolar macrophage subset has already been described and associated with a poor prognosis in various clinical entities. (IPF : Am J Respir Crit Care Med. 2019, PubMed: 31051082 ; COVID-19 : Cell 184, 6243-6261.e27 (2021)).

Response to Comment #5: In our revised manuscript, we have incorporated four new analyses testing for associations between the alveolar myeloid subsets and clinical parameters and biomarker profiles. Please refer to our Response to Comment #4 for more details as well as our response to Reviewer 1 Comment #1.

We have significantly expanded our discussion section to compare and contrast the subsets we have identified in our cohort of AHRF patients with other publications such as the ones you cite above. We agree the CD71-alveolar macrophage subset ("Mature Macrophages") in this cohort of AHRF patients is likely the same subset we (*JCI Insight* 2018;3:99281) and others (*AJRCCM*

2019;200:209) have identified in cohorts of patients with ARDS or IPF. Our study importantly links this phenotypic subset with transcriptional subsets described in scRNAseq databases (e.g. *AJRCCM* 2021;203:946, *Life Sci Alliance* 2022;5:e202201458) (described in Table S5). We argue the robust identification of this subset across different disease states is an important finding that strongly suggests “the replacement of mature alveolar macrophages with various monocyte and macrophage subsets is a common feature across many forms of lung injury.” We have incorporated the above discussion into the revised discussion section (paragraph #3).

Reviewer Comment #6: *Did the authors identify CD71 in their scRNA-seq analysis? It would be interesting to show features plot UMAPs with CD163 and CD71 expression.*

Response to Comment #6: We have included a UMAP of CD71 and CD163 cell-surface protein specificity in our revised manuscript (Figure S8) (which was also suggested by Reviewer #1). We agree this depiction of the data is a helpful complement to Figure 4.

Table S4 displays the most differentially expressed genes for each cluster. *CD163* (gene for CD163) was the 10th most differentially expressed marker gene for Cluster 0 (CD163/LGMN). *TFRC* (gene for CD71) was the 19th most differentially expressed marker gene for Cluster 3 (Mature Macrophages).

Reviewer Minor Comment #1: *Some abbreviations have not been introduced (e.g. LGMN, FCN1). The writing style could be reworked.*

Response to Minor Comment #1: We have ensured that all abbreviations are introduced in the manuscript. We have additionally formatted the entire manuscript to conform to the journal formatting requirements.

Reviewer Minor Comment #2: *There are some 'oversimplifications'. For example, overexpression of the CD86 marker is associated with the production of Th2 cytokines (discussion p 12), whereas many other studies associate this marker with 'M1' type polarization of alveolar macrophages (e.g. Plos one DOI: 10.1371/journal.pone.0045466; Nat. Immunol. DOI: 10.1038/ni.1937 ; Curr Opin Immunol DOI: 10.1016/j.coi.2010.01.009).*

Response to Minor Comment #2: We made major edits to the discussion section. We expanded the discussion of the most robust aspects of our paper (e.g. CD163, PD-L1) (discussion section paragraph #2 and #3), and eliminated the more speculative parts of the discussion (including the T_H2 discussion you specifically reference above).

Reviewer Minor Comment #3: *The discussion should be balanced and enriched by more references and comparisons with data in the literature.*

Response to Minor Comment #3: We made major edits to the results and discussion sections of the manuscript. We included a new Table S5 which contextualizes our findings within the published literature. Please refer to our responses to Reviewer #2 Minor Comment #1, Reviewer #3 Major Comment #3, Reviewer #3 Major Comment #5, and Reviewer #3 Minor

Comment #2 for more details on how we made our discussion more balanced and enriched by more references and comparisons with data in the literature.

Reviewer Minor Comment #4: *Analysis of developmental trajectory provides an indication, a bioinformatic assessment to infer the fate of a specific cell type but not a definitive demonstration of cell ontogeny.*

Response to Minor Comment #4: We agree the RNA velocity trajectory analysis is inferential of cell ontogeny and not definitive. We used very precise language in our results section to describe this analysis (e.g. “The developmental trajectory of a collection of individual cells can be **inferred** by the rate and direction of RNA velocity”, “These findings **suggest** that heterogenous alveolar monocyte/macrophage subsets converge into a transcriptionally distinct subset such as Intermediate Monocyte-Macrophages over the early course of AHRF that may repopulate the Mature AM subset.”).

We have also modified the limitations section our discussion section to explicitly state: “Fourth, we used RNA velocity trajectory analysis to infer cell ontogeny, however lineage tracing experiments are required to definitively demonstrate cell ontogeny” (discussion section paragraph #6).

Reviewer Minor Comment #5: *It would be interesting to discuss the role of CD71 and CD163 molecules.*

Response to Minor Comment #5: Thank you for emphasizing the importance of better contextualizing some of our study’s more robust findings. We have incorporated two new paragraphs in the discussion section to specifically discuss the role of CD71 and CD163 molecules (discussion section paragraph #2 and #3).

Reviewer Minor Comment #6: *The authors stated that it is not known whether alveolar monocyte and macrophage subsets that have been identified by scRNA-seq in subjects with COVID-19 are conserved in critically-ill patients without viral infection. Therefore, it would be interesting to compare the observations made in the present study with the results published for patients with COVID-19.*

Response to Minor Comment #6: Please refer to our new Table S5 and our responses to other comments above. Our revised manuscript specifically compares and contrasts our findings with those from studies that have enrolled patients with severe COVID-19.

REVIEWERS' COMMENTS

Reviewer #1 (Remarks to the Author):

My previous comments have been sufficiently addressed. The authors have provided an additional analysis on the relevance of their parsimonious marker set and on the specificity of the investigated cell-surface proteins. Importantly, they have extended the methods description and also included additional background information on how the QC thresholds for filtering cells were established.

I have two remaining minor points that the authors should address:

1. In Figure 2B and 2C multiple tests were performed and p-values < 0.05 were indicated as significant. However, from the description (both in figure caption and methods) it appears that no multiple-testing correction was used. The authors should apply a multiple-testing correction method and only indicate the significance level if the corrected p-value is below the threshold.

2. The plots in Figure 2B and 2C are extremely small and thus very difficult to read. The authors should only select a subset of plots for the main manuscript and include the remaining plots in the supplements, to allow for a larger plot size.

Reviewer #2 (Remarks to the Author):

The authors have addressed the comments in their revisions. Thank you.

Reviewer #3 (Remarks to the Author):

I thank the authors of this work for considering the criticisms made of this article. Overall, I find that they have provided complementary analyses and data that have improved the consistency of the results and their interpretation. I also appreciate the changes made to the discussion. In particular, I find that the conclusions are more modest and more in line with the data analysed. In summary, I still regret the lack of functional investigation of the

subsets identified and the partial validation of the surface markers identified on the basis of the combined single cell/cite-seq analysis. However, I am aware of the scarcity of samples and the inherent technical constraints. On the basis of these arguments and the modifications made, I am in favour of publishing this work.

Reviewer 1 General Comments: *My previous comments have been sufficiently addressed. The authors have provided an additional analysis on the relevance of their parsimonious marker set and on the specificity of the investigated cell-surface proteins. Importantly, they have extended the methods description and also included additional background information on how the QC thresholds for filtering cells were established.*

Response: Thank you very much for reviewing our manuscript. We think your suggestions significantly improved the quality and impact of the manuscript.

Reviewer Comment #1: *In Figure 2B and 2C multiple tests were performed and p-values < 0.05 were indicated as significant. However, from the description (both in figure caption and methods) it appears that no multiple-testing correction was used. The authors should apply a multiple-testing correction method and only indicate the significance level if the corrected p-value is below the threshold.*

Response to Comment #1: We having edited this figure (and legend) to include Bonferroni-adjusted p-values.

Reviewer Comment #2: *The plots in Figure 2B and 2C are extremely small and thus very difficult to read. The authors should only select a subset of plots for the main manuscript and include the remaining plots in the supplements, to allow for a larger plot size.*

Response to Comment #2: We edited Figure 2B and 2C to only include a subset of the plots that were most reflective of the different associations between subsets and clinical severity. We added a new Supplemental Figure S5 that includes all of the original plots (with Bonferroni-adjusted p-values as discussed above).

Reviewer 2 General Comments: *The authors have addressed the comments in their revisions. Thank you.*

Response: Thank you very much for reviewing our manuscript. We think your suggestions significantly improved the quality and impact of the manuscript.

Reviewer #3 General Comments: *I thank the authors of this work for considering the criticisms made of this article. Overall, I find that they have provided complementary analyses and data that have improved the consistency of the results and their interpretation. I also appreciate the changes made to the discussion. In particular, I find that the conclusions are more modest and more in line with the data analysed. In summary, I still regret the lack of functional investigation of the subsets identified and the partial validation of the surface markers identified on the basis of the combined single cell/cite-seq analysis. However, I am aware of the scarcity of samples and the inherent technical constraints. On the basis of these arguments and the modifications made, I am in favour of publishing this work.*

Response: Thank you very much for reviewing our manuscript. We think your suggestions significantly improved the quality and impact of the manuscript.